# Shelving, Stacking, Hanging: Relational Pose Diffusion for Multi-modal Rearrangement

**Anthony Simeonov**[1,3], **Ankit Goyal**[2,*], **Lucas Manuelli**[2,*], **Lin Yen-Chen**[1],
**Alina Sarmiento**[1,3], **Alberto Rodriguez**[1], **Pulkit Agrawal**[1,3,†], **Dieter Fox**[2,†]
[1]Massachusetts Institute of Technology    [2]NVIDIA    [3]Improbable AI Lab

**Abstract:** We propose a system for rearranging objects in a scene to achieve a desired object-scene placing relationship, such as a book inserted in an open slot of a bookshelf. The pipeline generalizes to novel geometries, poses, and layouts of both scenes and objects, and is trained from demonstrations to operate directly on 3D point clouds. Our system overcomes challenges associated with the existence of many geometrically-similar rearrangement solutions for a given scene. By leveraging an iterative pose de-noising training procedure, we can fit multi-modal demonstration data and produce multi-modal outputs while remaining precise and accurate. We also show the advantages of conditioning on relevant local geometric features while ignoring irrelevant global structure that harms both generalization and precision. We demonstrate our approach on three distinct rearrangement tasks that require handling multi-modality and generalization over object shape and pose in both simulation and the real world. Project website, code, and videos: https://anthonysimeonov.github.io/rpdiff-multi-modal

**Keywords:** Object Rearrangement, Multi-modality, Manipulation, Point Clouds

## 1 Introduction

Consider Figure 1, which illustrates (1) placing a book on a partially-filled shelf and (2) hanging a mug on one of the multiple racks on a table. These tasks involve reasoning about geometric interactions between an object and the scene to achieve a goal, which is a key requirement in many cleanup and de-cluttering tasks of interest to the robotics community [1]. In this work, we enable a robotic system to perform one important family of such tasks: 6-DoF rearrangement of rigid objects [2]. Our system uses point clouds obtained from depth cameras, allowing real-world operation with unknown 3D geometries. The rearrangement behavior is learned from a dataset of examples that show the desired object-scene relationship – a scene and (segmented) object point cloud are observed and a demonstrator transforms the object into a final configuration. For example, from a dataset showing books placed on shelves, our model learns how to transform new books into open shelf slots.

Real-world scenes are often composed of objects whose shapes and poses can vary independently. Such composition creates scenes that (i) present combinatorial variation in geometric appearance and layout (e.g., individual racks may be placed anywhere on a table) and (ii) offer many locations and geometric features for object-scene interaction (e.g., multiple slots for placing the book and multiple racks for hanging the mug). These features of real-world scenes bring about two key challenges for learning that go hand-in-hand: multi-modal placements and generalization to diverse scene layouts.

- **Multi-modality** appears in the rearrangement *outputs*. There may be many scene locations to place an object, and these multiple possibilities create difficulties during both learning and deployment. Namely, a well-known challenge in *learning* from demonstrations is fitting a dataset containing similar inputs that have different associated targets (modes). Moreover, during deployment, predicting multiple candidate rearrangements can help the robot choose the ones that also satisfy any additional constraints, such as workspace limits and collision avoidance. Therefore, the system must *predict* multi-modal outputs that span as many different rearrangement solutions as possible.

- **Generalization** must be addressed when processing the *inputs* to the system. A scene is composed of many elements that vary in both shape and layout. For example, a shelf can be located anywhere

---

Work done in part during NVIDIA internship, *Equal contribution, † Equal advising
7th Conference on Robot Learning (CoRL 2023), Atlanta, USA.

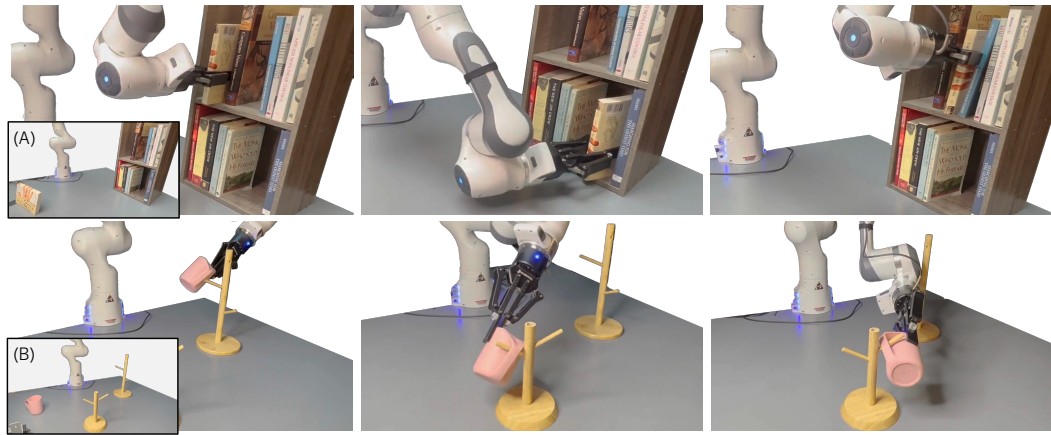

Figure 1: By learning from a set of demonstrations of a rearrangement task, such as *place the book in the shelf* (A) and *hang the mug on the rack* (B), Relational Pose Diffusion (RPDiff) can produce *multiple* transformations that achieve the same object-scene relationship for new object/scene pairs.

in the environment, and there are many possible arrangements of existing books within a shelf. The point clouds that are presented to the model reflect this diversity. Generalizing to such input variability is harder than generalizing to shape and pose variations for a single object, due to the combinatorially many arrangements and layouts of scenes.

Given a dataset of final object-scene point clouds (obtained by transforming the observed object point cloud into its resultant configuration at the end of the demo), we can synthesize many initial object configurations as perturbations of the final point clouds. Using this data, we can naturally cast rearrangement prediction as *point cloud pose de-noising*. From a final object-scene point cloud, we create a "noised" point cloud by randomly transforming the object and train a neural network to predict how to transform the noised point cloud back into the original configuration (using the known perturbation for ground truth supervision, see Fig. 2(b)). During deployment, we similarly predict a de-noising object transformation that satisfies the learned relation with the scene and use this predicted transformation as the rearrangement action. The robot executes the predicted rearrangement using a combination of grasp sampling, inverse kinematics, and motion planning.

Unfortunately, learning to de-noise from large perturbations (e.g., the "fully-noised" red point cloud in Fig. 2(b)) in one step can be ineffective when considering multi-modality [3] – creating similar-looking noised point clouds with prediction targets that differ can lead the model to learn an average solution that fits the data poorly. We overcome this difficulty by training the predictor as a diffusion model [4, 5] to perform *iterative* de-noising. By creating a *multi-step* noising process, diffusion models are trained to *incrementally* reverse the process one step at a time. Intuitively, early steps in this reverse process are closer to the ground truth and the associated prediction targets are more likely to be unique across samples – the prediction "looks more unimodal" to the model. The model similarly generates the test-time output in an iterative fashion. By starting this inference procedure from a diverse set of initial guesses, the predictions can converge to a diverse set of final solutions.

While iterative de-noising helps with multi-modality, we must consider how to support generalization to novel scene layouts. To achieve this, we propose to *locally encode* the scene point cloud by cropping a region near the object (e.g., see Fig. 2(c)). Locally cropping the input helps the model generalize by focusing on details in a local neighborhood and ignoring irrelevant and distant distractors. The features for representing smaller-scale patches can also be re-used across different spatial regions and scene instances [6–9]. We use a larger crop size on the initial iterations because the inference procedure starts from random guesses that may be far from a good solution. As the solution converges over multiple iterations, we gradually reduce the crop size to emphasize a more local scene context.

In summary, we present Relational Pose Diffusion (RPDiff), a method that performs 6-DoF relational rearrangement conditioned on an object and scene point cloud, that (1) generalizes across shapes, poses, and scene layouts, and (2) gracefully handles scenarios with multi-modality. We evaluate our approach in simulation and the real world on three tasks, (i) comparing to existing methods that either struggle with multi-modality and complex scenes or fail to achieve precise rearrangement, and (ii) ablating the various components of our overall pipeline.

**(A)** Evaluation: Starting from diverse initial poses, pose diffusion outputs a diverse set of rearrangement solutions

Input: Diverse initial poses $\left\{ \hat{\mathbf{T}}_k^{(I)} \right\}_{k=1}^{K}$

Output: Rearrangement transforms $\left\{ \hat{\mathbf{T}}_k^{(0)} \right\}_{k=1}^{K}$

**(B)** Training: Iterative pose de-noising for object-scene point cloud

Ground truth

$t = 1$  $t = 2$  $t = 3$  $t = T$

Noising (data gen)

$f_\theta$  $f_\theta$  De-noising (predictions)

**(C)** Network predicts SE(3) transform from object and cropped scene

$\bar{\mathbf{P}}_{\mathbf{S}}^{(t)} = \mathrm{crop}(\mathbf{P}_{\mathbf{S}}, \mathbf{P}_{\mathbf{O}}^{(t)})$

$\bar{\mathbf{P}}_{\mathbf{S}}^{(t)}$

$\mathbf{P}_{\mathbf{O}}^{(t)}$

$f_\theta$  $\hat{\mathbf{T}}_{\Delta}^{(t)}$

$\mathrm{pos\_emb}(t)$

$\mathbf{P}_{\mathbf{O}}^{(t-1)} = \hat{\mathbf{T}}_{\Delta}^{(t)} \mathbf{P}_{\mathbf{O}}^{(t)}$

Figure 2: **Method Overview.** (A) Starting from an object and scene point cloud $\mathbf{P}_{\mathbf{O}}$ and $\mathbf{P}_{\mathbf{S}}$, we transform $\mathbf{P}_{\mathbf{O}}$ to a diverse set of initial poses. RPDiff takes the initial object-scene point clouds as input, iteratively updates the object pose, and outputs a *set* of object configurations that satisfy a desired relationship with the scene. This enables integrating RPDiff with a planner to search for a placement to execute while satisfying additional system constraints. (B) The model is trained to perform *iterative pose de-noising*. Starting from object-scene point clouds that satisfy the desired task, we apply a sequence of perturbations to the object and train the model to predict SE(3) transforms that remove the noise one step at a time. (C) To facilitate generalization to novel scene layouts, we crop the scene point cloud to the region near the object point cloud.

## 2 Problem Setup

Our goal is to predict a set of SE(3) transformations $\{\mathbf{T}_k\}_{k=1}^{K}$ that accomplish an object rearrangement task given the scene $\mathbf{S}$ and the object $\mathbf{O}$, represented as 3D point clouds ($\mathbf{P}_{\mathbf{S}} \in \mathbb{R}^{M \times 3}$ and $\mathbf{P}_{\mathbf{O}} \in \mathbb{R}^{N \times 3}$, respectively). By selecting (i.e., via a learned scoring function) and applying one transformation from this set, we can place the object in a manner that fulfills the desired geometric relationship with the scene. We assume the object point cloud is segmented from the whole scene, which does not have any additional segmented objects (e.g., we cannot segment any individual books on the shelf). We also assume a training dataset $\mathcal{D} = \{(\mathbf{P}_{\mathbf{O}}, \mathbf{P}_{\mathbf{S}})\}_{l=1}^{L}$ where each data point represents an object placed at the desired configuration. For example, $\mathcal{D}$ could include point clouds of books and bookshelves (with different shapes, poses, and configurations of books on the shelf), and SE(3) transformations that place the books in one of the available slots. These demonstrations could come from a human or a scripted algorithm with access to ground truth object states in simulation.

Critically, depending on constraints imposed by other system components (e.g., available grasps, robot reachability, collision obstacles), the system must be capable of producing *multi-modal* output transformations. Predicting diverse outputs enables searching for a placement that can be feasibly executed. For execution on a robot, the robot has access to a grasp sampler [10], inverse kinematics (IK) solver, and motion planner to support generating and following a pick-and-place trajectory.

## 3 Method

The main idea is to iteratively de-noise the 6-DoF pose of the object until it satisfies the desired geometric relationship with the scene point cloud. An overview of our framework is given in Fig. 2.

### 3.1 Object-Scene Point Cloud Diffusion via Iterative Pose De-noising

We represent a rearrangement action $\mathbf{T}$ as the output of a multi-step de-noising process for a combined object-scene point cloud, indexed by discrete time variable $t = 0, ..., T$. This process reflects a transformation of the object point cloud in its initial noisy configuration $\mathbf{P}_{\mathbf{O}}^{(T)}$ to a final configuration $\mathbf{P}_{\mathbf{O}}^{(0)}$ that satisfies a desired relationship with the scene point cloud $\mathbf{P}_{\mathbf{S}}$, i.e., $\mathbf{P}_{\mathbf{O}}^{(0)} = \mathbf{T}\mathbf{P}_{\mathbf{O}}^{(T)}$. To achieve this, we train neural network $f_\theta : \mathbb{R}^{N \times 3} \times \mathbb{R}^{M \times 3} \to \mathrm{SE}(3)$ to predict an SE(3) transformation from the combined object-scene point cloud at each step. The network is trained as a diffusion model [4, 5] to incrementally reverse a manually constructed noising process that gradually perturbs the object point clouds until they match a distribution $\mathbf{P}_{\mathbf{O}}^{(T)} \sim p_{\mathbf{O}}^{(T)}(\cdot \mid \mathbf{P}_{\mathbf{S}})$, which we can efficiently sample from during deployment to begin de-noising at test time.

**Test-time Evaluation.** Starting with $\mathbf{P_O}$ and $\mathbf{P_S}$, we sample $K$ initial transforms $\{\hat{\mathbf{T}}_k^{(I)}\}_{k=1}^{K}$[*] and apply these to $\mathbf{P_O}$ to create initial object point clouds $\{\hat{\mathbf{P}}_{\mathbf{O},k}^{(I)}\}_{k=1}^{K}$ where $\hat{\mathbf{P}}_{\mathbf{O},k}^{(I)} = \hat{\mathbf{T}}_k^{(I)}\mathbf{P_O}$. For each of the $K$ initial transforms, we then perform the following update for $I$ steps.[†] At each iteration $i$:

$$\mathbf{T}_{\Delta}^{(i)} = \mathbf{T}_{\Delta}^{\text{Rand}} f_\theta\Big(\hat{\mathbf{P}}_{\mathbf{O}}^{(i)}, \mathbf{P_S}, \texttt{pos\_emb}(t)\Big) \qquad t = \texttt{i\_to\_t}(i) \tag{1}$$

$$\hat{\mathbf{T}}^{(i-1)} = \mathbf{T}_{\Delta}^{(i)}\hat{\mathbf{T}}^{(i)} \qquad \hat{\mathbf{P}}_{\mathbf{O}}^{(i-1)} = \mathbf{T}_{\Delta}^{(i)}\hat{\mathbf{P}}_{\mathbf{O}}^{(i)} \tag{2}$$

The update $\mathbf{T}_{\Delta}^{(i)}$ is formed by multiplying the denoising transform predicted by our model $f_\theta$ with a perturbation transform $\mathbf{T}_{\Delta}^{\text{Rand}}$ that is sampled from an iteration-conditioned normal distribution which converges toward deterministically producing an identify transform as $i$ tends toward 0. In the de-noising process, $\mathbf{T}_{\Delta}^{\text{Rand}}$ helps each of the $K$ samples converge to different multi-modal pose basins (analogously to the perturbation term in Stochastic Langevin Dynamics [11]). The function $\texttt{pos\_emb}$ represents a sinusoidal position embedding. Since $f_\theta$ is only trained on a finite set of $t$ values (i.e., $t = 1, ..., 5$) but we might want to perform the update in Eq. 2 for a larger number of steps, we use the function $\texttt{i\_to\_t}$ to map the iteration $i$ to a timestep value $t$ that the model has been trained on. Details on external noise scheduling and mapping $i$ to $t$ can be found in Appendix A3.

Generally, we search through $K$ solutions $\{\hat{\mathbf{T}}_k^{(0)}\}_{k=1}^{K}$ for one that can be executed while satisfying all other constraints (e.g., collision-free trajectory). However, we also want a way to select a single output to execute assuming there are no other constraints to satisfy. We may also want to reject "locally optimal" solutions that fail to complete the desired task. To achieve this, we use a separate classifier $h_\phi$ to score the predicted poses (i.e., $s_k = h_\phi(\mathbf{P_O}_k^{(0)}, \mathbf{P_S})$ where $s \in [0,1]$), such that the sample indexed with $k_{\text{exec}} = \texttt{argmax}\ \{s_k\}_{k=1}^{K}$ can be selected for execution[‡].

**Training.** Given a dataset sample $(\mathbf{P_O}, \mathbf{P_S})$, we start with final "placed" object point cloud $\mathbf{P_O}^{(0)} = \mathbf{P_O}$ and randomly sampled timestep $t \in [1, T]$. We then obtain a perturbation transform $\mathbf{T}_{\text{noise}}^{(t)}$ from a timestep-conditioned distribution with appropriately scaled variance and create a noised point cloud $\mathbf{P_O}^{(t)} = \mathbf{T}_{\text{noise}}^{(t)}\mathbf{P_O}$. The task is to predict a transformation that takes one de-noising step as $\hat{\mathbf{T}}_{\Delta}^{(t)} = f_\theta(\mathbf{P_O}^{(t)}, \mathbf{P_S}, \texttt{pos\_emb}(t))$. Network parameters $\theta$ are trained to minimize a loss between the prediction $\hat{\mathbf{T}}_{\Delta}^{(t)}$ and a ground truth target $\mathbf{T}_{\Delta,\text{GT}}^{(t)}$. We use the Chamfer distance between the point cloud obtained by applying the predicted transform and the ground-truth next point cloud as the loss to minimize.

A natural target for $f_\theta$ to predict is the inverse of the perturbation, i.e., $\mathbf{T}_{\Delta,\text{GT}}^{(t)} = \mathbf{T}_{\text{noise,inv}}^{(t)} = \big[\mathbf{T}_{\text{noise}}^{(t)}\big]^{-1}$, to encourage recovering the original sample. However, as the perturbation magnitude varies across timesteps, this requires output predictions of different scales for different timesteps. In supervised learning with neural networks, it is advisable to keep the magnitudes of both input and output signals consistent in order to minimize large fluctuations in gradient magnitudes between samples [12]. For this reason, an alternative approach is to encourage the network to take shorter "unit steps" in the *direction* of the original sample. We achieve this by uniformly interpolating the full inverse perturbation as $\{\mathbf{T}_{\text{interp}}^{(s)}\}_{s=1}^{t} = \texttt{interp}(\mathbf{T}_{\text{noise, inv}}^{(t)}, t)$ and training the network to predict one interval in this interpolated set, i.e., $\mathbf{T}_{\Delta,\text{GT}}^{(t)} = [\mathbf{T}_{\text{interp}}^{(t-1)}]^{-1}\mathbf{T}_{\text{interp}}^{(t)}$ (details in Appendix A2 and A7).

For the success classifier, we generate positive and negative rearrangement examples, where positives use the final demonstration point cloud, $\mathbf{P_O}^{(0)}$, and negatives are obtained by sampling diverse perturbations of $\mathbf{P_O}^{(0)}$. The classifier weights $\phi$ (separate from weights $\theta$) are trained to minimize a binary cross-entropy loss between the predicted likelihood and the ground truth success labels.

## 3.2 Architecture

We use a Transformer [13] for processing point clouds and making pose predictions. A Transformer is a natural architecture for both (i) identifying important geometric parts *within* the object and the scene and (ii) capturing relationships that occur *between* the important parts of the object and the

---

[*] Initial rotations are drawn from a uniform grid over $SO(3)$ , and we uniformly sample translations that position the object within the bounding box of the scene point cloud.

[†] We denote application of $SE(3)$ transform $\mathbf{T} = (\mathbf{R}, \mathbf{t})$ to 3D point $\mathbf{x}$ as $\mathbf{T}\mathbf{x} = \mathbf{R}\mathbf{x} + \mathbf{t}$

[‡] See Appendix A7 for results showing that scoring with $h_\phi$ performs better than, e.g., uniform output sampling

scene. Starting with $\mathbf{P_O}$ and $\mathbf{P_S}$, we tokenize the point clouds to obtain input features. This can be performed by passing through a point cloud encoder [14, 15], but we simply downsample the point clouds and use the downsampled 3D point features as input. We then pass these input tokens through a Transformer encoder and decoder, which performs self-attention on the scene point cloud, and cross-attention between the scene and the object. This produces output features for each point, which are mean-pooled to obtain a global feature vector. The global feature is passed to a set of MLPs which predict the rotation $\mathbf{R} \in \mathrm{SO}(3)$ and a translation $\mathbf{t} \in \mathbb{R}^3$. As in [10, 16], we represent the rotation by predicting vectors $a \in \mathbb{R}^3$ and $b \in \mathbb{R}^3$, finding the component of $b$ that is orthogonal to $a$, and normalizing to obtain $\hat{a}$ and $\hat{b}$. We then take a cross product to obtain $\hat{c} = \hat{a} \times \hat{b}$, and construct $\mathbf{R}$ as $\begin{bmatrix} \hat{a} & \hat{b} & \hat{c} \end{bmatrix}$. We incorporate iteration $t$ by passing $\mathrm{pos\_emb}(t)$ as a global token in the decoder and adding it to the global output feature. To predict success likelihood, we process point clouds with the same Transformer but output a single scalar followed by a sigmoid.

### 3.3 Local Conditioning

The approach described above conditions the transform regression on both the object and the scene. However, distant global information can act as a distraction and hamper both precision and generalization. Prior work has also observed this and suggested hard attention mechanisms on the input observation like cropping task-relevant regions to improve generalization by ignoring irrelevant distractors [8, 9]. Building on this intuition, we modify the scene point cloud by cropping $\mathbf{P_S}$ to only include points that are near the current object point cloud $\mathbf{P_O}^{(i)}$. Our modified pose prediction thus becomes $\hat{\mathbf{T}}_\Delta^{(i)} = f_\theta\left(\hat{\mathbf{P}}_\mathbf{O}^{(i)}, \bar{\mathbf{P}}_\mathbf{S}^{(i)}, \mathrm{pos\_emb}(\mathtt{i\_to\_t}(i))\right)$ where $\bar{\mathbf{P}}_\mathbf{S}^{(i)} = \mathrm{crop}(\hat{\mathbf{P}}_\mathbf{O}^{(i)}, \mathbf{P_S})$. The function $\mathrm{crop}$ returns the points in $\mathbf{P_S}$ that are within an axis-aligned box centered at the mean of $\hat{\mathbf{P}}_\mathbf{O}^{(i)}$. We try one variant of the $\mathrm{crop}$ function that returns a fixed-size crop, and another that adjusts the crop size depending on the iteration variable $i$ (the size starts large and gradually decreases for later iterations).

## 4 Experiments: Design and Setup

Our quantitative experiments in simulation are designed to answer the following questions:

1. How well does RPDiff achieve the desired tasks compared to other methods for rearrangement?
2. How successful is RPDiff in producing a diverse set of transformations compared to baselines?
3. How does our performance change with different components modified or removed?

We also demonstrate RPDiff within a pick-and-place pipeline in the real world to further highlight the benefits of multi-modal generation and our ability to transfer from simulation to the real world.

### 4.1 Task Descriptions and Training Data Generation

We evaluate our method on three tasks that emphasize multiple available object placements: (1) placing a book on a partially-filled bookshelf, (2) stacking a can on a stack of cans or an open shelf region, and, (3) hanging a mug on one of many racks with many hooks. As a sanity check for our baseline implementations, we also include two easier versions of "mug on rack" tasks that are "less multi-modal". These consist of (i) hanging a mug on one rack with a single hook and (ii) hanging a mug on one rack with two hooks. We programmatically generate ~1k-3k demonstrations of each task in simulation with a diverse set of procedurally generated shapes (details in Appendix A2). We use each respective dataset to train both RPDiff and each baseline (one model for each task). For our real-world experiments, we directly transfer and deploy the models trained on simulated data.

### 4.2 Evaluation Environment Setup

**Simulation.** We conduct quantitative experiments in the PyBullet [17] simulation engine. The predicted transform is applied to the object by simulating an insertion controller which directly actuates the object's center of mass (i.e., there is no robot in the simulator). The insertion is executed from a "pre-placement" pose that is offset from the predicted placement. This offset is obtained using prior knowledge about the task and the objects and is not predicted (see Appendix A6 for details). To quantify performance, we report the success rate over 100 trials, using the final simulator state to compute success. We also quantify coverage by comparing the set of predictions to a ground truth set of feasible solutions and computing the corresponding precision and recall. Details on the insertion controller, computation of $\mathbf{T}^{\mathrm{pre\text{-}place}}$, and the task success criteria can be found in the Appendix.

| Method | Mug/EasyRack | Mug/MedRack | Book/Shelf | Mug/Multi-MedRack | Can/Cabinet |
|---|---|---|---|---|---|
| C2F Q-attn | 0.31 | 0.31 | 0.57 | 0.26 | 0.51 |
| R-NDF-base | 0.75 | 0.29 | 0.00 | 0.00 | 0.14 |
| NSM-base | 0.83 | 0.17 | 0.02 | 0.01 | 0.08 |
| NSM-base + CVAE | – | 0.39 | 0.17 | 0.27 | 0.19 |
| RPDiff (**ours**) | **0.92** | **0.83** | **0.94** | **0.86** | **0.85** |

Table 1: **Rearrangement success rates in simulation.** On tasks with a unimodal solution space and simpler scene geometry, each method performs well (see **Mug/EasyRack** task). However, on tasks involving more significant shape variation and multi-modality, RPDiff works better than all other approaches.

**Real World.** We also apply RPDiff to object rearrangement in the real world using a Franka Panda robotic arm with a Robotiq 2F140 parallel jaw gripper. We use four calibrated depth cameras to observe the tabletop environment. From the cameras, we obtain point clouds $\mathbf{P_O}$ and $\mathbf{P_S}$ of object $\mathbf{O}$ and scene $\mathbf{S}$ and apply our method to predict transformation $\mathbf{T}$. $\mathbf{T}$ is applied to $\mathbf{O}$ by transforming an initial grasp pose $\mathbf{T}_{\text{grasp}}$ (using a separate grasp predictor [10]) by $\mathbf{T}$ to obtain a placing pose $\mathbf{T}_{\text{place}} = \mathbf{TT}_{\text{grasp}}$, and inverse kinematics and motion planning is used to reach $\mathbf{T}_{\text{grasp}}$ and $\mathbf{T}_{\text{place}}$.

### 4.3 Baselines

**Coarse-to-Fine Q-attention (C2F-QA).** This method adapts the classification-based approach proposed in [8] to relational rearrangement. We train a fully convolutional network to predict a distribution of scores over a voxelized representation of the scene, denoting a heatmap over candidate translations of the object centroid. The model runs in a "coarse-to-fine" fashion by performing this operation multiple times over a smaller volume at higher resolutions. On the last step, we pool the voxel features and predict a distribution over a discrete set of rotations to apply to the object. We use our success classifier to rank the predicted transforms and execute the output with the top score.

**Relational Neural Descriptor Fields (R-NDF).** R-NDF [18] uses a neural field shape representation trained on category-level 3D models as a feature space wherein local coordinate frames can be matched via nearest-neighbor search. R-NDFs have been used to perform relational rearrangement tasks via the process of encoding and localizing task-relevant coordinate frames near the object parts that must align to achieve the desired rearrangement. We call this method "R-NDF-base" because it does not feature the additional energy-based model for refinement proposed in the original work.

**Neural Shape Mating (NSM) + CVAE.** Neural Shape Mating (NSM) [3] uses a Transformer to process a pair of point clouds and predict how to align them. Architecturally, NSM is the same as our relative pose regression model, with the key differences of (i) being trained on arbitrarily large perturbations of the demonstration point clouds, (ii) not using local cropping, and (iii) only making a single prediction. We call this baseline "NSM-base" because we do not consider the auxiliary signed-distance prediction and learned discriminator proposed in the original approach [3]. While the method performs well on unimodal tasks, the approach is not designed to handle multi-modality. Therefore, we modify NSM to act as a conditional variational autoencoder (CVAE) [19] to better enable learning from multi-modal data. We use NSM+CVAE to predict multiple transforms and execute the output with the top score produced by our success classifier.

## 5 Results

### 5.1 Simulation: Success Rate Evaluation

Table 1 shows the success rates achieved by each method on each task and highlights that our method performs best across the board. The primary failure mode from C2F-QA is low precision in the rotation prediction. Qualitatively, the C2F-QA failures are often close to a successful placement but still cause the insertion to fail. In contrast, our refinement procedure outputs very small rotations that can precisely align the object relative to the scene.

Similarly, we find R-NDF performs poorly on more complex scenes with many available placements. We hypothesize this is because R-NDF encodes scene point clouds into a global latent representation. Since the single set of latent variables must capture all possible configurations of the individual scene components, global encodings fail to represent larger-scale scenes with significant geometric variability [6, 7]. For instance, R-NDF can perform well with individual racks that all have a single hook, but fails when presented with multiple racks.

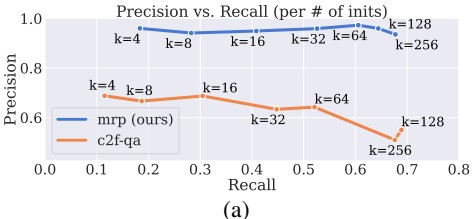

| Crop Method | Success Rate | | |
|---|---|---|---|
| | Mug/Rack | Book/Shelf | Can/Cabinet |
| None | 0.58 | 0.62 | 0.42 |
| Fixed | 0.76 | 0.92 | 0.75 |
| Varying | 0.86 | 0.94 | 0.85 |

| | |
|:---:|:---:|
| (a) | (b) |

Figure 3: (a) **Coverage evaluation in simulation.** Both RPDiff and C2F-QA achieve high placement coverage, but the prediction quality of C2F-QA reduces with an increase in coverage, while RPDiff produces outputs that remain precise while achieving high coverage. (b) **Cropping ablations.** Success rate of RPDiff with different kinds of scene point cloud conditioning. The increased success rate achieved when using local scene cropping highlights the generalization and precision benefits of focusing on a local spatial region.

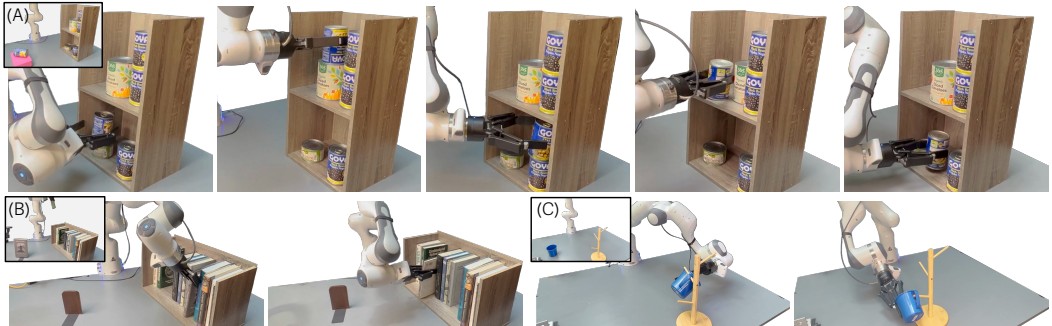

Figure 4: **Real-world multi-modal rearrangement.** Executing **Can/Cabinet** (A), **Book/Shelf** (B), and **Mug/Rack** (C) in the real world. For each task, the initial object-scene configuration is shown in the top-left image, and examples of executing multiple inferred placements are shown in the main image sequence.

Finally, while NSM+CVAE improves upon the unimodal version of NSM, we find the generated transforms vary too smoothly between the discrete modes (e.g., book poses that lie in between the available shelf slots), an effect analogous to the typical limitation of VAE-based generators producing blurry outputs in image generation. We hypothesize this over-smoothing is caused by trying to make the approximate posterior match the unimodal Gaussian prior. This contrasts RPDiff's ability to "snap on" to the available placing locations in a given scene. More discussion on the performance obtained by the baseline methods and how they are implemented can be found in Appendix A6.

## 5.2 Simulation: Coverage Evaluation

Next, we evaluate the ability to produce multi-modal outputs that cover the space of rearrangement solutions and examine the tradeoff between prediction quality and coverage. Since coverage is affected by the number of parallel runs we perform, we compute average recall and average precision for different values of $K$ (the number of initial poses that are refined). Precision and recall are computed with respect to a set of ground truth rearrangement solutions for a given object-scene instance. We consider positive predictions as those that are within a 3.5cm position and 5-degree rotation threshold of a ground truth solution.

Fig. 3a shows the results for our approach along with C2F-QA, the best-performing baseline. We observe a trend of better coverage (higher recall) with more outputs for both approaches. For a modest value of $K = 32$, we observe RPDiff is able to cover over half of the available placement solutions on average, with C2F-QA achieving slightly lower coverage. However, we see a stark difference between the methods in terms of precision as the number of outputs is increased. C2F-QA suffers from more outputs being far away from any ground truth solution, while our approach maintains consistently high generation quality even when outputting upwards of 200 rearrangement poses.

## 5.3 Simulation: Local Cropping Ablations and Modifications

Finally, we evaluate the benefits of introducing local scene conditioning into our relative pose regression model. Table 3b shows the performance variation of our method with different kinds of scene point cloud conditioning. We achieve the best performance with the version of local conditioning that varies the crop sizes on a per-iteration basis. Using a fixed crop size marginally

reduces performance, while conditioning on the whole uncropped scene point cloud performs much worse. This highlights the generalization and precision benefits of focusing on a local spatial region near the object in its imagined configuration. It also suggests an advantage of using a coarse-to-fine approach that considers a larger region on earlier iterations. Additional results examining the effect of the success classifier, external noise, and parameterization of i_to_t can be found in Appendix A7.

### 5.4 Real World: Object rearrangement via pick-and-place

Finally, we use RPDiff to perform relational rearrangement via pick-and-place on real-world objects and scenes. Fig. 1 and Fig. 4 show the robot executing *multiple* inferred placements on our three tasks. We relied on our approach's ability to output multiple solutions, as some geometrically valid placements were not kinematically feasible for the robot based on its workspace limits and the surrounding collision geometry. Please see the supplemental video for real-world execution.

## 6 Related Work

**Object Rearrangement from Perception**. Object rearrangement that operates with unknown objects in the real world by operating from perceptual input has been an area of growing interest [2, 3, 8, 18, 20–46]. One straightforward method is end-to-end training to directly regress the relative transformation, as in Neural Shape Mating (NSM) [3]. Others have explored identifying task-relevant object parts and then solving for the desired alignment, as in TAX-Pose and R-NDF [18, 37, 45]. However, many of these approaches in their naive form struggle when there is *multi-modality* (NSM and TAX-Pose can only output a single solution). There has been success addressing multi-modality by performing classification over a discretized version of the search space [8, 39, 41, 43, 44, 47], but these methods are typically less precise.

**Denoising Diffusion and Iterative Regression**. Diffusion models [4, 48] use an iterative de-noising process to perform generative modeling. While they were originally designed for generating images, they have been extended to other domains including waveforms [49, 50], 3D shapes [51, 52], and decision-making[53–55]. Several approaches have applied diffusion models (and related energy-based models) to a variety of robotics domains, including policy learning [56, 57], motion planning/trajectory optimization [58–60], grasping [54], and object rearrangement [18, 38, 61]. The use of iterative regression has also been successful in other domains such as pose estimation [62–65].

SE(3)-DiffusionFields [54] integrate learned 6-DoF grasp distributions within a trajectory optimization framework, and LEGO-Net [55] employs iterative de-noising to generate realistic-looking room layouts. Our work differs in that we do not assume known object states or 3D models. Most similar to our work, StructDiffusion [38] uses a diffusion model to perform language-conditioned object rearrangement with point clouds. While the focus in [38] is to rearrange multiple objects into abstract structures (e.g., circles, lines) specified via natural language, we emphasize covering all rearrangement modes and integrating with sampling-based planners.

## 7 Limitations and Conclusion

**Limitations**. The amount of demonstration data we use may be difficult to obtain in the real world, thus we rely on scripted policies that use privileged information in simulation for demo collection. Furthermore, sim2real distribution shifts reduce our real-world performance, we lack a closed-loop control policy for placement execution that is robust to perturbations, and we do not show any transfer to new tasks. Finally, a subtle yet important limitation is our use of manually-computed pre-placement offset poses. Predicting the final desired object configuration is an important step toward general-purpose rearrangement, but it would be even better to also predict additional waypoint transforms that help obtain a feasible *path* to the final pose.

**Conclusion**. This work presents an approach for rearranging objects in a scene to achieve a desired placing relationship, while operating with novel geometries, poses, and scene layouts. Our system can produce multi-modal distributions of object transformations for rearrangement, overcoming the difficulty of fitting multi-modal demonstration datasets and facilitating integration with planning algorithms that require diverse actions to search through. Our results illustrate the capabilities of our framework across a diverse range of rearrangement tasks involving objects and scenes that present a large number of feasible rearrangement solutions.

# 8 Acknowledgement

The authors would like to thank NVIDIA Seattle Robotics Lab members and the MIT Improbable AI Lab for their valuable feedback and support in developing this project. In particular, we would like to acknowledge Idan Shenfeld, Anurag Ajay, and Antonia Bronars for helpful suggestions on improving the clarity of the draft. This work was partly supported by Sony Research Awards and Amazon Research Awards. Anthony Simeonov is supported in part by the NSF Graduate Research Fellowship.

**Author Contributions**

**Anthony Simeonov** conceived the overall project goals, investigated several approaches for addressing multi-modality in rearrangement prediction, implemented the pose diffusion framework, wrote all the code, ran simulation and real-world experiments, and was the primary author of the paper.

**Ankit Goyal** advised the project, made technical suggestions on clarifying the method and improving the experimental evaluation, supported iteration on obtaining real robot results, and helped with writing the paper.

**Lucas Manuelli** engaged in research discussions about rearrangement prediction, suggested initial ideas for addressing multi-modality, advised the project in its early stages, and provided valuable feedback on the paper.

**Lin Yen-Chen** supported early project brainstorming, helped develop direct connections with diffusion models, gave feedback on evaluation tasks, and helped edit the paper.

**Alina Sarmiento** helped implement the framework on the real robot and implemented the grasp generation model that enabled the pick-and-place demos on the Franka Panda.

**Alberto Rodriguez** engaged in technical discussions on the connections to iterative optimization methods and integrating the framework in the context of a sampling-based planner.

**Pulkit Agrawal** suggested connections to work on iterative regression that came before diffusion models, helped clarify key technical insights on the benefits of iterative prediction, suggested ablations, helped with paper writing/editing, and co-advised the project.

**Dieter Fox** was involved in technical discussions on relational tasks involving object part interactions, proposed some of the evaluation tasks, helped formalize connections to other related work, and advised and supported the overall project.

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

# Shelving, Stacking, Hanging: Relational Pose Diffusion for Multi-modal Rearrangement – Supplementary Material

Section A1 includes additional visualizations of iterative test-time evaluation on simulated shapes and examples of object-scene point clouds that were used as training data. In Section A2, we present details on data generation, model architecture, and training for RPDiff. In Section A3 we elaborate in more detail on the multi-step iterative regression inference procedure which predicts the set of rearrangement transforms. Section A4 describes more details about how the success classifier is trained and used in conjunction with our transform predictor as a simple mechanism for selecting which among multiple candidate transforms to execute. In Section A5, we describe more details about our experimental setup, and Section A6 discusses more details on the evaluation tasks, robot execution pipelines, and methods used for computing pre-placement offset poses. In Section A7 we present an additional set of ablations to highlight the impact of other hyperparameters and design decisions. Section A8 describes additional implementation details for the real-world executions along with an expanded discussion on limitations and avenues for future work. Section A9 includes preliminary results on training a multi-task model for iterative pose de-noising and using RPDiff to perform multi-step manipulation, and Section A10 includes additional discussion on demo collection (and the manually-designed heuristics it uses), performance analysis and sim2real considerations, system engineering details, expanded related works. Finally, Section A11 shows model architecture diagrams a summarized set of relevant hyperparameters that were used in training and evaluation.

## A1    Additional Test-time and Training Data Visualizations

Here, we show additional visualizations of the tasks used in our simulation experiments and the noised point clouds used to train our pose regression model. Figure A1 shows snapshots of performing the iterative de-noising at evaluation time with simulated objects, and Figure A2 shows examples of the combined object-scene point clouds and their corresponding noised versions that were used for training to perform iterative de-noising.

## A2    Iterative Pose Regression Training and Data Generation

This section describes the data used for training our pose diffusion model, the network architecture we used for processing point clouds and predicting SE(3) transforms, and details on training.

### A2.1    Training Data Generation

**Objects used in simulated rearrangement demonstrations**. We create the rearrangement demonstrations in simulation with a set of synthetic 3D objects. The three tasks we consider include objects from five categories: mugs, racks, cans, books, "bookshelves" (shelves partially filled with books), and "cabinets" (shelves partially-filled with stacks of cans). We use ShapeNet [66] for the mugs and procedurally generate our own dataset of .obj files for the racks, books, shelves, and cabinets. See Figure A3 for representative samples of the 3D models from each category.

**Procedurally generated rearrangement demonstrations in simulation**.  The core regression model $f_\theta$ in RPDiff is trained to process a combined object-scene point cloud and predict an SE(3) transformation updates the pose of the object point cloud. To train the model to make these relative pose predictions, we use a dataset of demonstrations showing object and scene point clouds in final configurations that satisfy the desired rearrangement tasks. Here we describe how we obtain these "final point cloud" demonstrations

We begin by initializing the objects on a table in PyBullet [17] in random positions and orientations and render depth images with the object segmented from the background using multiple simulated cameras. These depth maps are converted to 3D point clouds and fused into the world coordinate frame using known camera poses. To obtain a diverse set of point clouds, we randomize the number of cameras (1-4), camera viewing angles, distances between the cameras and objects, object scales, and object poses. Rendering point clouds in this way allows the model to see some of the occlusion patterns that occur when the objects are in different orientations and cannot be viewed from below the table. To see enough of the shelf/cabinet region, we use the known state of the shelf/cabinet to position two cameras that roughly point toward the open side of the shelf/cabinet.

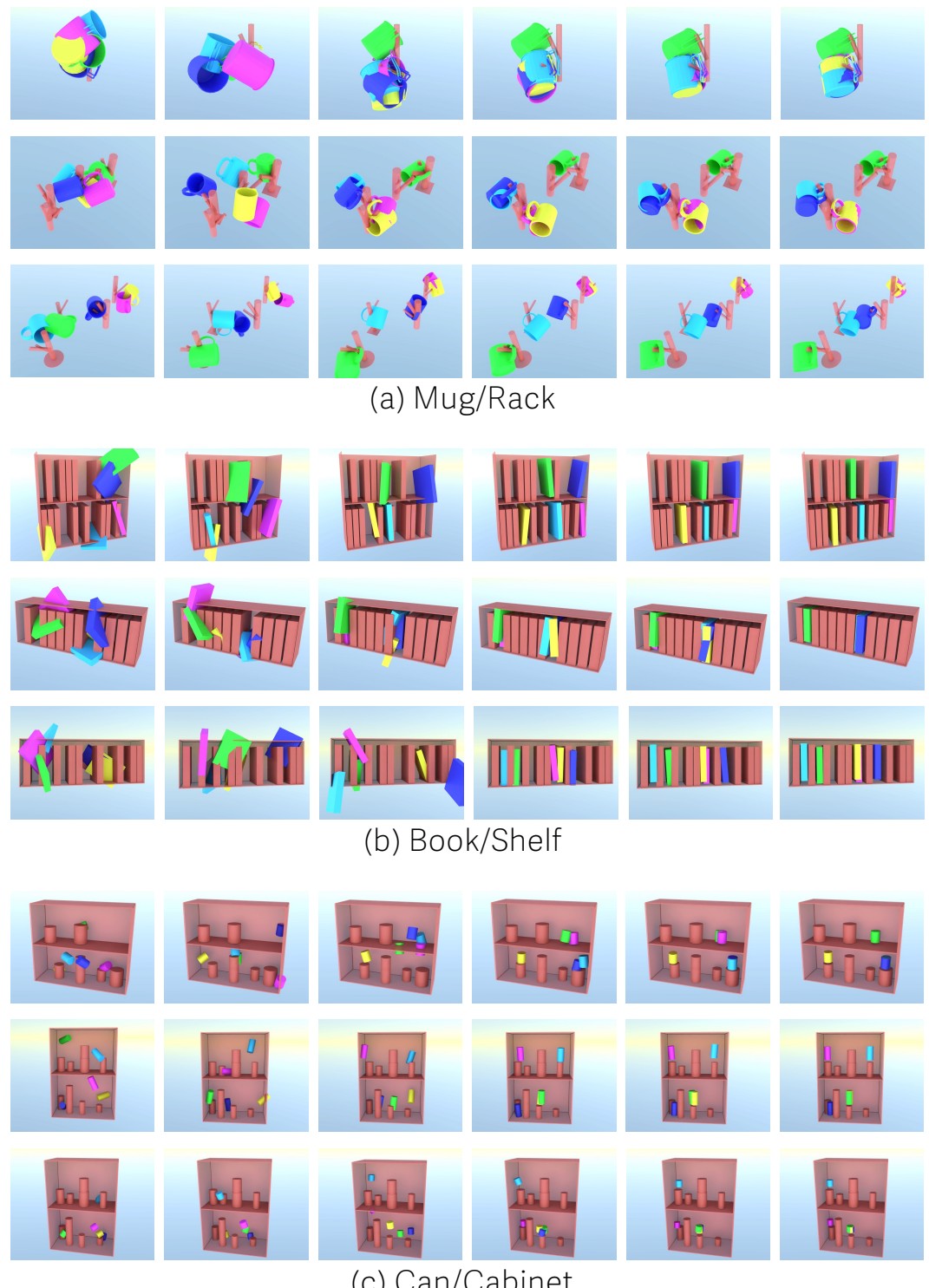

(a) Mug/Rack

(b) Book/Shelf

(c) Can/Cabinet

Figure A1: Visualizations of multiple steps of iterative de-noising on simulated objects. Starting from the left side, each object is initialized in a random SE(3) pose in the vicinity of the scene. Over multiple iterations, RPDiff updates the object pose. The right side shows the final set of converged solutions.

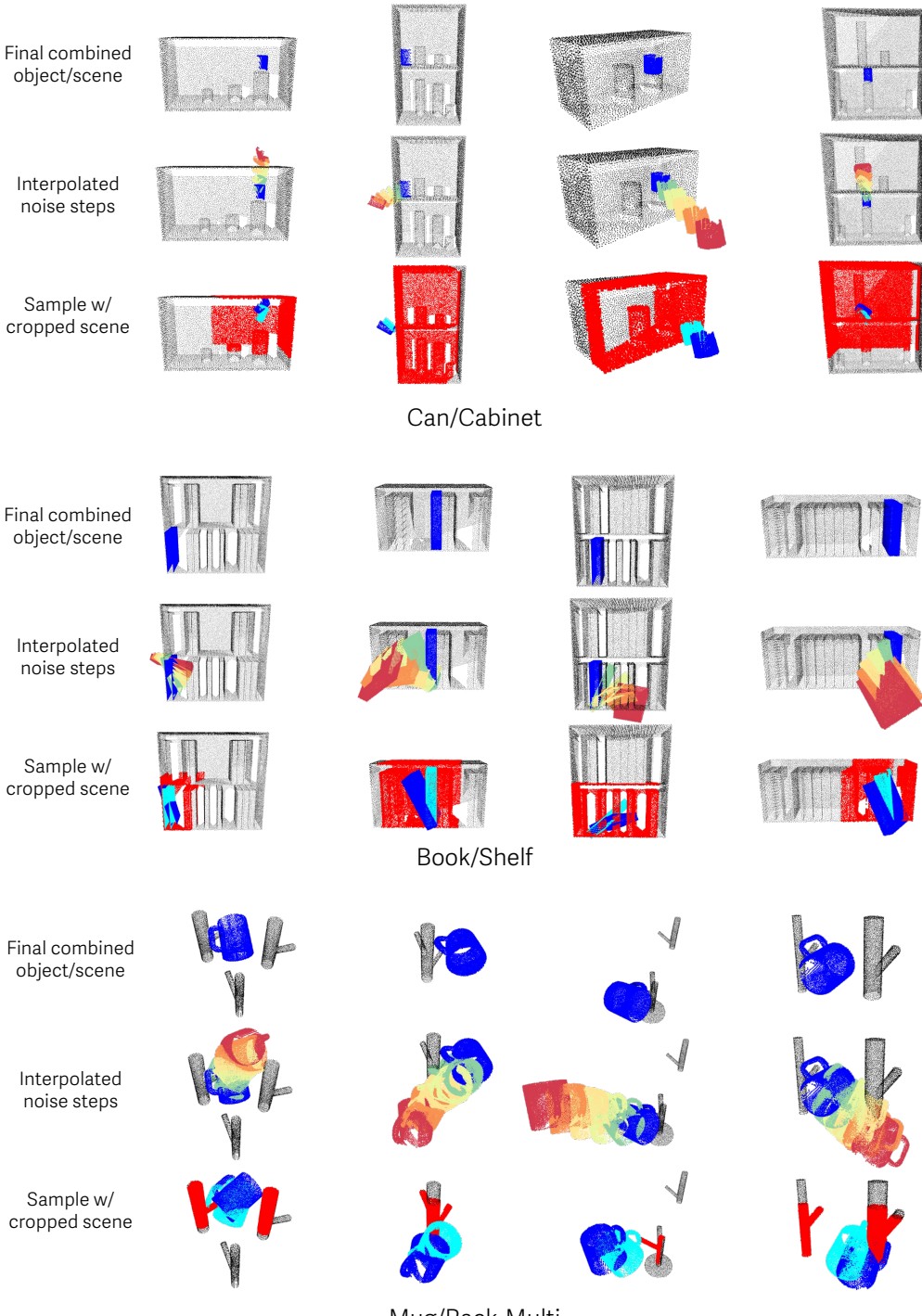

Figure A2: Example point clouds from the demonstrations for each task of **Can/Cabinet** (top), **Book/Shelf** (middle) and **Mug/RackMed-Multi** (bottom). For each task, the top row shows the ground truth combined object-scene point cloud. Scene point clouds are in black and object point clouds are in dark blue. The middle row in each task shows an example of creating multiple steps of noising perturbations by uniformly interpolating a single randomly sampled perturbation transform (with a combination of linear interpolation for the translation and SLERP for the rotation). Different colors show the point clouds at different interpolated poses. The bottom row shows a sampled step among these interpolated poses, with the corresponding "noised" object point cloud (dark blue), ground truth target point cloud (light blue), and cropped scene point cloud (red).

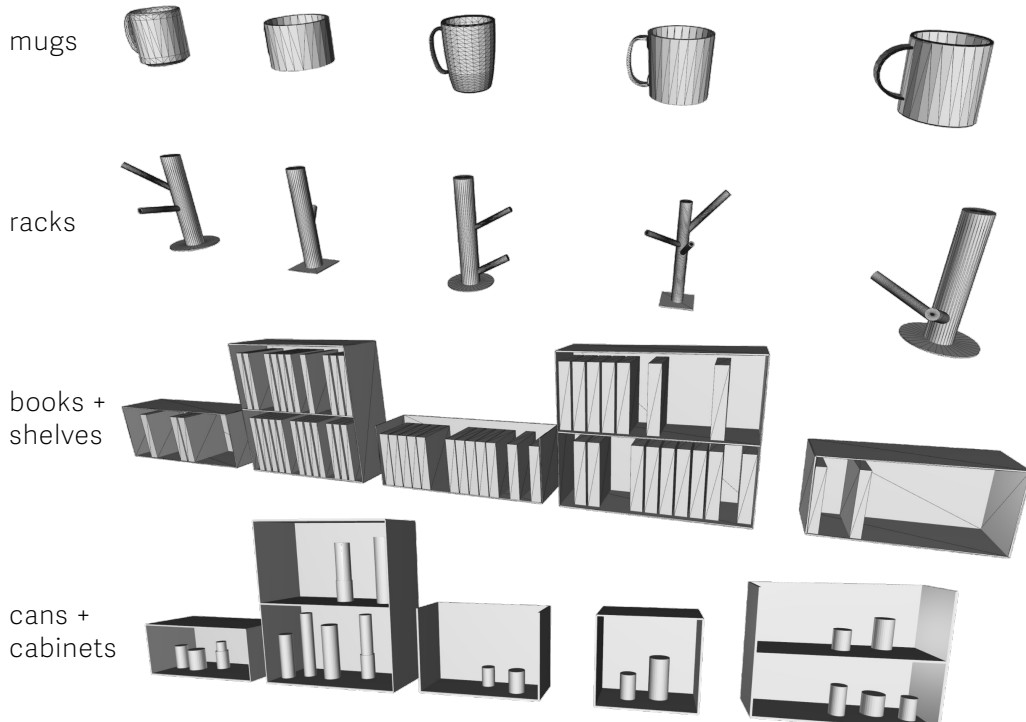

Figure A3: Example 3D models used to train RPDiff and deploy RPDiff on our rearrangement tasks. Mugs are from ShapeNet [66] while we procedurally generated our own synthetic racks, books, cans, shelves, and cabinets.

After obtaining the initial object and scene point clouds, we obtain an SE(3) transform to apply to the object, such that transforming into a "final" objct pose using this transform results in the desired placement. This transform is used to translate and rotate the initial object point cloud, such that the combined "final object" and scene point cloud can be used for generating training examples. Figure A2 shows example visualizations of the final point clouds in the demonstrations for each task.

We obtain the final configuration that satisfies these tasks using a combination of privileged knowledge about the objects in the simulator (e.g., ground truth state, approximate locations of task-relevant object parts, 3D mesh models for each object, known placing locations that are available) and human intuition about the task. To create mug configurations that satisfy "hanging" on one of the pegs of a rack, we first approximately locate one of the pegs on one of the racks (we select one uniformly at random) and the handle on the mug (which is straightforward because all the ShapeNet mugs are aligned with the handle pointing in the +y axis of the body frame). We then transform the mug so that the handle is approximately "on" the selected hook. Finally, we sample small perturbations about this nominal pose until we find one that does not lead to any collision/penetration between the two shapes. We perform an analogous process for the other tasks, where the ground truth available slots in the bookshelf and positions that work for placing the mug (e.g., on top of a stack, or on a flat shelf region in between existing stacks) are recorded when the 3D models for the shelves/cabinets are created. The exact methods for generating these shapes and their corresponding rearrangement poses can be found in our code.

## A2.2 Pose Prediction Architecture

**Transformer point cloud processing and pose regression**. We follow the Transformer [13] architecture proposed in Neural Shape Mating [3] for processing point clouds and computing shape features that are fed to the output MLPs for pose prediction.

We first downsample the observed point clouds $\mathbf{P_O} \in \mathbb{R}^{N' \times 3}$ and $\mathbf{P_S} \in \mathbb{R}^{M' \times 3}$ using farthest point sampling into $\bar{\mathbf{P}}_\mathbf{O} \in \mathbb{R}^{N \times 3}$ and $\bar{\mathbf{P}}_\mathbf{S} \in \mathbb{R}^{M \times 3}$. We then normalize to create $\mathbf{P_O}^{\text{norm}} \in \mathbb{R}^{N \times 3}$ and $\mathbf{P_S}^{\text{norm}} \in \mathbb{R}^{M \times 3}$, based on the centroid of the scene point cloud and a scaling factor that approximately scales the combined point cloud to have extents similar to a unit bounding box:

$$\bar{\mathbf{P}}_\mathbf{S} = \begin{bmatrix} \mathbf{p}_1^\mathbf{S} \\ \mathbf{p}_2^\mathbf{S} \\ \dots \\ \mathbf{p}_M^\mathbf{S} \end{bmatrix} \qquad \bar{\mathbf{P}}_\mathbf{O} = \begin{bmatrix} \mathbf{p}_1^\mathbf{O} \\ \mathbf{p}_2^\mathbf{O} \\ \dots \\ \mathbf{p}_M^\mathbf{O} \end{bmatrix} \qquad \mathbf{p}^{\mathbf{S},\text{cent}} = \frac{1}{M} \sum_{i=1}^{M} \mathbf{p}_i^\mathbf{S} \qquad \mathbf{a} = \max\{\mathbf{p}_i^\mathbf{S}\} - \min\{\mathbf{p}_i^\mathbf{S}\}$$

$$\mathbf{P_S}^{\text{norm}} = \begin{bmatrix} \mathbf{p}_1^{\mathbf{S},\text{norm}} \\ \mathbf{p}_2^{\mathbf{S},\text{norm}} \\ \dots \\ \mathbf{p}_M^{\mathbf{S},\text{norm}} \end{bmatrix} \qquad \mathbf{p}_i^{\mathbf{S},\text{norm}} = \mathbf{a}(\mathbf{p}_i^\mathbf{S} - \mathbf{p}^{\mathbf{S},\text{cent}}) \quad \forall\, i \in 1, ..., M$$

$$\mathbf{P_O}^{\text{norm}} = \begin{bmatrix} \mathbf{p}_1^{\mathbf{O},\text{norm}} \\ \mathbf{p}_2^{\mathbf{O},\text{norm}} \\ \dots \\ \mathbf{p}_M^{\mathbf{O},\text{norm}} \end{bmatrix} \qquad \mathbf{p}_i^{\mathbf{O},\text{norm}} = \mathbf{a}(\mathbf{p}_i^\mathbf{O} - \mathbf{p}^{\mathbf{S},\text{cent}}) \quad \forall\, j \in 1, ..., N$$

Next, we "tokenize" the normalized object/scene point clouds into $d$-dimensional input features $\phi_\mathbf{O} \in \mathbb{R}^{N \times d}$ and $\phi_\mathbf{S} \in \mathbb{R}^{M \times d}$. We directly use the 3D coordinate features from the downsampled and normalized point clouds as input tokens, and then project the input to a $d$-dimensional vector with a linear layer $\mathbf{W}_{\text{in}} \in \mathbb{R}^{d \times 3}$:

$$\phi_\mathbf{S} = \begin{bmatrix} \mathbf{W}_{\text{in}}\mathbf{p}_1^{\mathbf{S},\text{norm}} \\ \mathbf{W}_{\text{in}}\mathbf{p}_2^{\mathbf{S},\text{norm}} \\ \dots \\ \mathbf{W}_{\text{in}}\mathbf{p}_M^{\mathbf{S},\text{norm}} \end{bmatrix} \qquad \phi_\mathbf{O} = \begin{bmatrix} \mathbf{W}_{\text{in}}\mathbf{p}_1^{\mathbf{O},\text{norm}} \\ \mathbf{W}_{\text{in}}\mathbf{p}_2^{\mathbf{O},\text{norm}} \\ \dots \\ \mathbf{W}_{\text{in}}\mathbf{p}_M^{\mathbf{O},\text{norm}} \end{bmatrix}$$

Note we could also pass the point cloud through a point cloud encoder to pool local features together, as performed in NSM via DGCNN [15]. We did not experiment with this as we obtained satisfactory results by directly operating on the individual point features, but it would likely perform similarly or even better if we first passed through a point cloud encoder. We also incorporate the timestep $t$ that the current prediction corresponds to by including the position-encoded timestep as an additional input token together with the object point tokens as $\bar{\phi}_\mathbf{O} \in \mathbb{R}^{(N+1) \times d}$ where $\bar{\phi}_\mathbf{O} = \begin{bmatrix} \phi_\mathbf{O} \\ \texttt{pos\_emb}(t) \end{bmatrix}$.

We then use a Transformer encoder and decoder to process the combined tokenized point cloud (see Figure A4 for visual depiction). This consists of performing multiple rounds of self-attention on the scene features (encoder) and then performing a combination of self-attention on the object point cloud together with cross-attention between the object point cloud and the output features of the scene point cloud (decoder):

$$q_\mathbf{S} = Q_E(\phi_\mathbf{S}) \quad k_\mathbf{S} = K_E(\phi_\mathbf{S}) \quad v_\mathbf{S} = V_E(\phi_\mathbf{S})$$

$$s_\mathbf{S} = \text{Attention}(q_\mathbf{S}, k_\mathbf{S}, v_\mathbf{S}) = \text{softmax}\left(\frac{q_\mathbf{S} k_\mathbf{S}^\mathsf{T}}{\sqrt{d}}\right) v_\mathbf{S}$$

$$q_\mathbf{O} = Q_D(\bar{\phi}_\mathbf{O}) \quad k_\mathbf{O} = K_D(\bar{\phi}_\mathbf{O}) \quad v_\mathbf{O} = V_D(\bar{\phi}_\mathbf{O})$$

$$s_\mathbf{O} = \text{Attention}(q_\mathbf{O}, k_\mathbf{O}, v_\mathbf{O}) = \text{softmax}\left(\frac{q_\mathbf{O} k_\mathbf{O}^\mathsf{T}}{\sqrt{d}}\right) v_\mathbf{O}$$

$$h_\mathbf{O} = \text{Attention}(q = s_\mathbf{O}, k = s_\mathbf{S}, v = s_\mathbf{S}) = \text{softmax}\left(\frac{s_\mathbf{O} s_\mathbf{S}^\mathsf{T}}{\sqrt{d}}\right) s_\mathbf{S}$$

This gives a set of output features $h_\mathbf{O} \in \mathbb{R}^{(N+1) \times d}$ where $d$ is the dimension of the embedding space. We compute a global feature by mean-pooling the output point features and averaging with the timestep embedding as a residual connection, and then use a set of output MLPs to predict the translation and rotation (the rotation is obtained by converting a pair of 3D vectors into an orthonormal

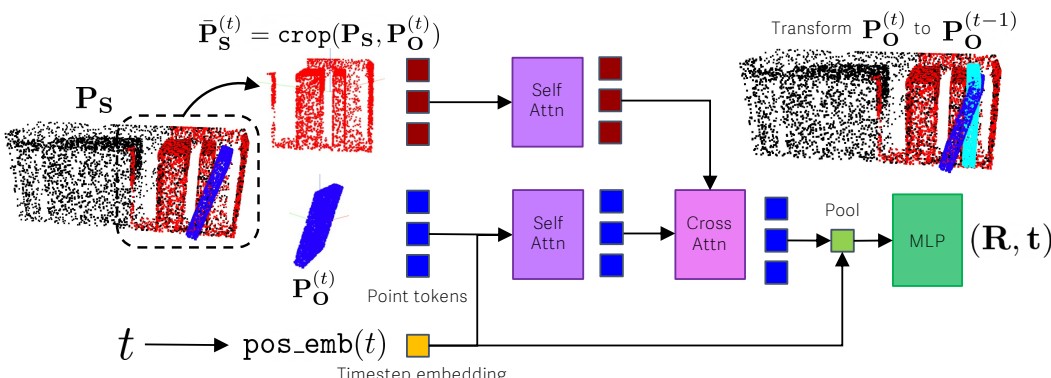

Figure A4: Architecture diagram showing a combination of self-attention and cross-attention among object and scene point cloud for SE(3) transform prediction. The scene point cloud is processed via multiple rounds of self-attention, while the object features are combined via a combination of self-attention and cross-attention with the scene point cloud. The timestep embedding is incorporated as both an input token and via a residual connection with the pooled output feature. The global output feature is used to predict the translation and rotation that are applied to the object point cloud.

basis and then stacking into a rotation matrix [10, 16]):

$$\bar{h}_{\mathbf{O}} = \frac{1}{2}\Big(\frac{1}{N}\sum_{i=1}^{N+1} h_{\mathbf{O},i} + \texttt{pos\_emb}(t)\Big) \qquad\qquad \bar{h}_{\mathbf{O}} \in \mathbb{R}^d$$

$$\mathbf{t} = \text{MLP}_{\text{trans}}(\bar{h}_{\mathbf{O}}) \qquad\qquad \mathbf{t} \in \mathbb{R}^3$$

$$a,\ b = \text{MLP}_{\text{rot}}(\bar{h}_{\mathbf{O}}) \qquad\qquad a \in \mathbb{R}^3,\ b \in \mathbb{R}^3$$

$$\hat{a} = \frac{a}{||a||} \qquad \hat{b} = \frac{b - \langle\hat{a}, b\rangle\hat{a}}{||b||} \qquad \hat{c} = \hat{a} \times \hat{b}$$

$$\mathbf{R} = \begin{bmatrix} | & | & | \\ \hat{a} & \hat{b} & \hat{c} \\ | & | & | \end{bmatrix}$$

**Local scene point cloud cropping**. As shown in the experimental results, local cropping helps improve performance due to increasing precision while generalizing well to unseen layouts of the scene. Our "Fixed" cropping method uses a box with fixed side length $L_{\text{box}} = L_{\text{min}}$, centered at the current object point cloud iterate across all timesteps, and selects scene point cloud points that lie within this box. Our "Varying" cropping method adjusts the length of the box based on the timestep, with larger timesteps using a larger crop, and smaller timesteps using a smaller crop. We parameterize this as a function of the timestep $t$ via the following linear decay function:

$$L_{\text{box}} = L_{\text{min}} + (L_{\text{max}} - L_{\text{min}})\frac{T - t}{T}$$

where $L_{\text{min}}$ and $L_{\text{max}}$ are hyperparameters.

**Applying Predicted Transforms to Object Point Cloud**. We apply the predicted rotation and translation by first mean-centering the object point cloud, applying the rotation, and then translating back to the original world frame position, and then finally translating by the predicted translation. This helps reduce sensitivity to the rotation prediction, whereas if we rotate about the world frame coordinate axes, a small rotation can cause a large configuration change in the object.

### A2.3 Training Details

Here we elaborate on details regarding training the RPDiff pose diffusion model using the demonstration data and model architecture described in the sections above. A dataset sample consists of a tuple $(\mathbf{P}_{\mathbf{O}}, \mathbf{P}_{\mathbf{S}})$. From this tuple, we want to construct a perturbed object point cloud $\mathbf{P}_{\mathbf{O}}^{(t)}$ for a particular timestep $t \in 1, ..., T$, where lower values of $t$ correspond to noised point clouds that are more similar

| Skill Type | Number of samples |
|---|---:|
| **Mug/EasyRack** | 3190 |
| **Mug/MedRack** | 950 |
| **Mug/Multi-MedRack** | 3240 |
| **Book/Shelf** | 1720 |
| **Can/Cabinet** | 2790 |

Table 2: Number of demonstrations used in each task. The same set of demonstrations is used to train both our method and each baseline method.

to the ground truth, and larger values of $T$ are more perturbed. At the limit, the distribution of point clouds corresponding to $t = T$ should approximately match the distribution we will sample from when initializing the iterative refinement procedure at test time.

Noising schedules and perturbation schemes are an active area of research currently in the diffusion modeling litierature [67, 68], and there are many options available for applying noise to the data samples. We apply a simple method that makes use of uniformly interpolated SE(3) transforms. First, we sample one "large" transform from the same distribution we use to initialize the test-time evaluation procedure from – rotations are sampled uniformly from SO(3) and translations are sampled uniformly within a bounding box around the scene point cloud. We then use a combination of linear interpolation on the translations, and spherical-linear interpolation (SLERP) on the rotations, to obtain a sequence of $T$ uniformly-spaced transforms (see Fig. A2 for example visualizations). Based on the sampled timestep $t$, we select the transform corresponding to timestep $t$ in this sequence as the noising perturbation $\mathbf{T}_{\text{noise}}^{(t)}$, and use the transform corresponding to timestep $t - 1$ to compute the "incremental"/"interval" transform to use as a prediction target. As discussed in Section 3.1, using the incremental transform as a prediction target helps maintain a more uniform output scale among the predictions across samples, which is beneficial for neural network optimization as it minimizes gradient fluctuations [12]. We also provide quantitative evidence that predicting only the increment instead of the full inverse perturbation benefits overall performance. See Section A7 for details.

The main hyperparameter for this procedure is the number of steps $T$. In our experiments, we observed it is important to find an appropriate value for $T$. When $T$ is too large, the magnitude of the transforms between consecutive timesteps is very small, and the iterative predictions at evaluation time make tiny updates to the point cloud pose, oftentimes failing to converge. When $T$ is too small, most of the noised point clouds will be very far from the ground truth and might look similar across training samples but require conflicting prediction targets, which causes the model to fit the data poorly. We found that values in the vicinity of $T = 5$ work well across our tasks ($T = 2$ and $T = 50$ both did not work well). This corresponds to an average perturbation magnitude of 2.5cm for the translation and 18 degrees for the rotation.

After obtaining the ground truth prediction target, we compute the gradient with respect to the loss between the prediction and the ground truth, which is composed of the Chamfer distance between the point cloud obtained by applying the predicted transform and the ground truth next point cloud. We also found the model to work well using combined translation mean-squared error and geodesic rotation distance [69, 70] loss.

We trained a separate model for each task, with each model training for 500 thousand iterations on a single NVIDIA V100 GPU with a batch size of 16. We used a learning rate schedule of linear warmup and cosine decay, with a maximum learning rate of 1e-4. Training takes about three days. We train the models using the AdamW [71] optimizer. Table 2 includes the number of demonstrations we used for each task.

## A3  Test time evaluation

Here, we elaborate in more detail on the iterative de-noising procedure performed at test time. Starting with $\mathbf{P_O}$ and $\mathbf{P_S}$, we sample $K$ initial transforms $\{\hat{\mathbf{T}}_k^{(I)}\}_{k=1}^{K}$, where initial rotations are drawn from a uniform grid over SO(3) , and we uniformly sample translations that position the object within the

bounding box of the scene point cloud. We create $K$ copies of $\mathbf{P_O}$ and apply each corresponding transform to create initial object point clouds $\{\hat{\mathbf{P}}_{\mathbf{O},k}^{(I)}\}_{k=1}^{K}$ where $\hat{\mathbf{P}}_{\mathbf{O},k}^{(I)} = \hat{\mathbf{T}}_k^{(I)} \mathbf{P_O}$. We then perform the following update for $I$ steps for each of the $K$ initial transforms:

$$\hat{\mathbf{T}}^{(i-1)} = (\mathbf{T}_{\Delta}^{\text{Rand}} \hat{\mathbf{T}}_{\Delta}) \hat{\mathbf{T}}^{(n)} \qquad \hat{\mathbf{P}}_{\mathbf{O}}^{(n-1)} = (\mathbf{T}_{\Delta}^{\text{Rand}} \hat{\mathbf{T}}_{\Delta}) \hat{\mathbf{P}}_{\mathbf{O}}^{(i)}$$

where transform $\hat{\mathbf{T}}_{\Delta}$ is obtained as $\hat{\mathbf{T}}_{\Delta} = f_{\theta}(\hat{\mathbf{P}}_{\mathbf{O}}^{(i)}, \mathbf{P_S}, \texttt{pos\_emb}(\texttt{i\_to\_t}(i)))$. Transform $\mathbf{T}_{\Delta}^{\text{Rand}}$ is sampled from a timestep-conditioned uniform distribution that converges toward deterministically producing an identify transform as $i$ tends toward 0. We obtain the random noise by sampling from a Gaussian distribution for both translation and rotation. For the translation, we directly output a 3D vector with random elements. For the rotation, we represent the random noise via axis angle 3D rotation $\mathbf{R}_{\text{aa}}^0 \in \mathbb{R}^3$ and convert it to a rotation matrix using the SO(3) exponential map [72] (and a 3D translation $\mathbf{t}^0 \in \mathbb{R}^3$). We exponentially decay the variance of these noise distributions so that they produce nearly zero effect as the iterations tend toward 0. We perform the updates in a batch. The full iterative inference procedure can be found in Alg. 1.

**Evaluation timestep scheduling and prediction behavior for different timestep values..** The function $\texttt{i\_to\_t}$ is used to map the iteration number $i$ to a timestep value $t$ that the model has been trained on. This allows the number of steps during evaluation ($I$) to differ from the number of steps during training ($T$). For example, we found values of $T = 5$ to work well during training but used a default value of $I = 50$ for evaluation. We observed this benefits performance since running the iterative evaluation procedure for many steps helps convergence and enables "bouncing out" of "locally optimal" solutions. However, we found that if we provide values for $i$ that go beyond the support of what the model is trained on (i.e., for $i > T$), the predictions perform poorly. Thus, the function $\texttt{i\_to\_t}$ ensures all values $i \in 1, ..., I$ are mapped to an appropriate value $t \in 1, ..., T$ that the model has seen previously.

There are many ways to obtain this mapping, and different implementations produce different kinds of behavior. This is because different $\texttt{i\_to\_t}$ schedules emphasize using the model in different ways since the model learns qualitatively different behavior for different values of $t$. Specifically, for smaller values of $t$, the model has only been trained on "small basins of attraction" and thus the predictions are more precise and local, which allows the model to "snap on" to any solution in the immediate vicinity of the current object iterate. Figure A5 shows this in a set of artifically constrained evaluation runs where the model is constrained to use the *same* timestep for every step $i = 1, ..., I$.

However, this can also lead the model to get stuck near regions that are far from any solution. On the other hand, for larger perturbations, the data starts to look more multi-modal and the model averages out toward either a biased solution in the direction of a biased region, or just an identity transform that doesn't move the object at all.

We find the pipeline performs best when primarily using predictions corresponding to smaller timesteps, but still incorporating predictions from higher timesteps. We thus parameterize the timestep schedule $\texttt{i\_to\_t}$ such that it exponentially increases the number of predictions used for smaller values of $t$. While there are many ways this can be implemented, we use the following procedure: we construct an array $D$ of length $I$ where each element lies between 1 and $T$, and define the mapping $\texttt{i\_to\_t}$ as

$$t = \texttt{i\_to\_t}(i) = D_i \qquad \text{subscript } i \text{ denotes the } i\text{-th element of } D$$

The array $D$ is parameterized by a constant value $A$ (where higher value of $A$ corresponds to using more predictions with smaller timesteps, while $A = 1$ corresponds to using each timestep an equal number of times) and ensures that predictions for each timestep are made at least once:

**Algorithm 1** Rearrangement Transform Inference via Iterative Point Cloud De-noising

---

1: **Input:** Scene point cloud $\mathbf{P_S}$, object point cloud $\mathbf{P_O}$, number of parallel runs $K$, number of iterations to use in evaluation $I$, number of iterations used in training $T$, pose regression model $f_\theta$, success classifier $h_\phi$, function to map from evaluation iteration values to training iteration values `i_to_t`, parameters for controlling what fraction of evaluation iterations correspond to smaller training timestep values $A$, local cropping function `crop`, distribution for sampling external pose noise $p_{\text{AnnealedRandSE(3)}}$

    # Init transforms, transformed object, and cropped scene
2: **for** $k$ in 1,...,$K$ **do**
3:     $\mathbf{R}_k^{(H)} \sim p_{\text{UnifSO(3)}}(\cdot)$
4:     $\mathbf{t}_k^{(H)} \sim p_{\text{UnifBoundingBox}}(\cdot \mid \mathbf{P_O}, \mathbf{P_S})$
5:     $\hat{\mathbf{T}}_k^{(H)} = \begin{bmatrix} \mathbf{R} & \mathbf{t} \\ \mathbf{0} & 1 \end{bmatrix}$
6:     $\hat{\mathbf{P}}_{\mathbf{O},k}^{(H)} = \hat{\mathbf{T}}_k^{(H)} \mathbf{P_O}$
7:     $\bar{\mathbf{P}}_{\mathbf{S},k}^{(H)} = \texttt{crop}(\hat{\mathbf{P}}_{\mathbf{O},k}^{(H)}, \mathbf{P_S})$
8: **end for**
    # Init set of transform and final point cloud solutions and classifier scores
9: `init` $\mathcal{S} = \emptyset$
10: `init` $\mathcal{T} = \emptyset$
11: `init` $\mathcal{P} = \emptyset$
    # Iterative pose regression
12: **for** $i$ in $I$,...,1 **do**
      # Map evaluation timestep to in-distribution training timestep
13:     $t = \texttt{i\_to\_t}(i, A)$
14:     **for** $k$ in 1,...,$K$ **do**
15:         $\hat{\mathbf{T}}_{\Delta,k} = f_\theta(\mathbf{P}_{\mathbf{O},k}^{(t)}, \bar{\mathbf{P}}_{\mathbf{S},k}^{(t)}, \texttt{pos\_emb}(t))$
16:         **if** $i > (0.2 * I)$ **then**
          # Apply random external noise, with noise magnitude annealed as $i$ approaches 0
17:           $\mathbf{T}_{\Delta,k}^{\text{Rand}} \sim p_{\text{AnnealedRandSE(3)}}(\cdot \mid i)$
18:         **else**
          # Remove all external noise for the last 20% of the iterations
19:           $\mathbf{T}_{\Delta,k}^{\text{Rand}} = \mathbf{I}_4$
20:         **end if**
21:         $\hat{\mathbf{T}}_k^{(i-1)} = \mathbf{T}_{\Delta,k}^{\text{Rand}} \mathbf{T}_{\Delta,k} \hat{\mathbf{T}}_k^{(i)}$
22:         $\hat{\mathbf{P}}_{\mathbf{O},k}^{(i-1)} = \mathbf{T}_{\Delta,k}^{\text{Rand}} \mathbf{T}_{\Delta,k} \hat{\mathbf{P}}_{\mathbf{O},k}^{(i)}$
23:         $\bar{\mathbf{P}}_{\mathbf{S},k}^{(i-1)} = \texttt{crop}(\hat{\mathbf{P}}_{\mathbf{O},k}^{(i-1)}, \mathbf{P_S}, t, T)$
24:         **if** $i == 1$ **then**
          # Predict success probabilities from final objects
25:           $s_k = h_\phi(\mathbf{P}_{\mathbf{O},k}^{(0)}, \mathbf{P_S})$
          # Save final rearrangement solutions and predicted scores
26:           $\mathcal{S} = \mathcal{S} \cup \{s_k\}$
27:           $\mathcal{T} = \mathcal{T} \cup \{\hat{\mathbf{T}}_k^{(0)}\}$
28:           $\mathcal{P} = \mathcal{T} \cup \{\hat{\mathbf{P}}_{\mathbf{O},k}^{(0)}\}$
29:         **end if**
30:     **end for**
31: **end for**
    # Decision rule (e.g., argmax) for output
32: $k^{\text{out}} = \texttt{argmax}(\mathcal{S})$
33: $\mathbf{T}^{\text{out}} = \mathcal{T}[k^{\text{out}}]$
    # Return top-scoring transform and full set of solutions for potential downstream planning/search
34: **return** $\mathbf{T}^{\text{out}}, \mathcal{T}, \mathcal{P}, \mathcal{S}$

---

$f_\theta(\mathbf{P_O}, \mathbf{P_S}, \texttt{pos\_emb}(1))$

$f_\theta(\mathbf{P_O}, \mathbf{P_S}, \texttt{pos\_emb}(2))$

$f_\theta(\mathbf{P_O}, \mathbf{P_S}, \texttt{pos\_emb}(3))$

$f_\theta(\mathbf{P_O}, \mathbf{P_S}, \texttt{pos\_emb}(4))$

$f_\theta(\mathbf{P_O}, \mathbf{P_S}, \texttt{pos\_emb}(5))$

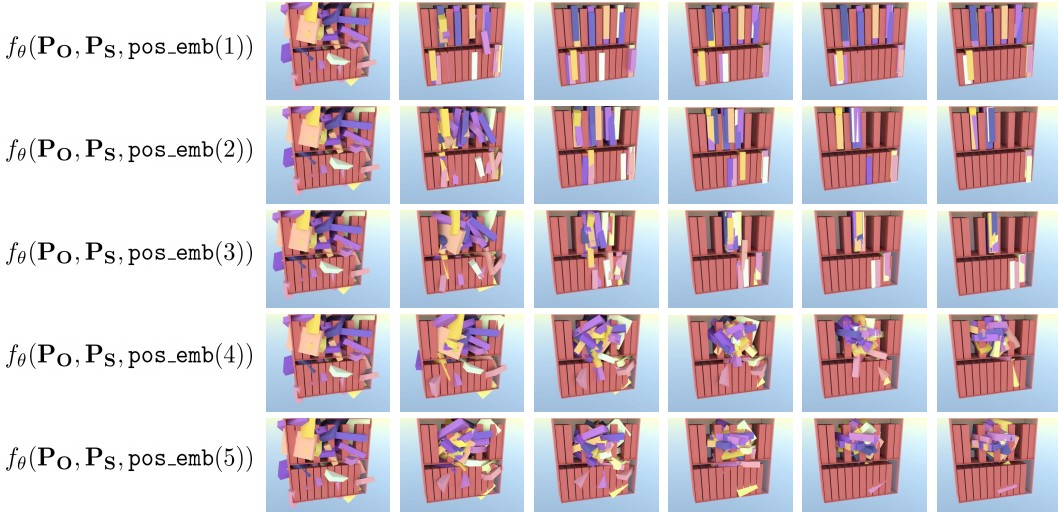

Figure A5: Examples of running our full iterative evaluation procedure for $I$ steps with the model constrained to use a fixed value for $t$ on each iteration. This highlights the different behavior the model has learned for different timesteps in the de-noising process. For timesteps near 1, the model has learned to make very local updates that "snap on" to whatever features are in the immediate vicinity of the object. As the timesteps get larger, the model considers a more global context and makes predictions that reach solutions that are farther away from the initial object pose. However, these end up more biased to a single solution in a region where there may be many nearby solutions (see the top row of shelves where there are four slots that the model finds when using timestep 1, but the model only reaches two of them with timestep $t = 2$ and one of them with $t = 3$). For even larger values of $t$, the model has learned a much more biased and "averaged out" solution that fails to rotate the object and only approximately reaches the scene regions corresponding to valid placements.

$$B = [A^T, , A^{T-1}..., A^2, A^1] \qquad \text{Exponentially decreasing values}$$

$$C = \lceil \frac{A * I}{\sum_{i=1}^{T} A_i} \rceil \qquad \text{Normalize, scale up by } I, \text{ and round up (minimum value per element is 1)}$$

$$\bar{C} = \lceil \frac{C * I}{\sum_{i=1}^{T} C_i} \rceil \qquad \text{Normalize again so } \sum_{i=1}^{T} \bar{C}_i \approx I \text{ with } \bar{C}_i \in \mathbb{N} \, \forall \, i = 1, ..., T$$

$$\bar{C}_1 = \bar{C}_0 - (\sum_{i=1}^{T} \bar{C}_i - I) \qquad \text{Ensure } \sum_{i=1}^{T} \bar{C}_i = I \text{ exactly}$$

Then, from $\bar{C}$, we construct multiple arrays with values ranging from 1 to $T$, each with lengths corresponding to values in $\bar{C}$,

$$\bar{D}_1 = [\bar{D}_{1,1} \ \bar{D}_{1,2} \ ...] \text{ with } \bar{D}_{1,k} = 1 \, \forall k \in 1, ..., \bar{C}_1$$
$$\bar{D}_2 = [\bar{D}_{2,1} \ \bar{D}_{2,2} \ ...] \text{ with } \bar{D}_{2,k} = 2 \, \forall k \in 1, ..., \bar{C}_2$$
$$...$$
$$\bar{D}_T = [\bar{D}_{T,1} \ \bar{D}_{T,2} \ ...] \text{ with } \bar{D}_{T,k} = T \, \forall k \in 1, ..., \bar{C}_T$$

and then stack these arrays together to obtain $D$ as a complete array of length $I$:

$$D = [\bar{D}_1 \ \bar{D}_2 \ ... \ \bar{D}_T]$$

## A4  Success Classifier Details

In this section, we present details on training and applying the success classifier $h_\phi$ that we use for ranking and filtering the set of multiple predicted SE(3) transforms produced by RPDiff.

**Training Data**. To train the success classifier, we use the demonstrations to generate positive and negative examples, where the positives are labeled with success likelihood 1.0 and the negatives have success likelihood 0.0. The positives are simply the unperturbed final point clouds and the negatives are perturbations of the final object point clouds. We use the same sampling scheme of sampling a rotation from a uniform distribution over SO(3) and sampling a translation uniformly from within a bounding box around the scene point cloud.

**Model Architecture**. We use an identical Transformer architecture as described in Section A2, except that we use a single output MLP followed by a sigmoid to output the predicted success likelihood, we do not condition on the timestep, and we provide the uncropped scene point cloud.

**Training Details**. We supervise the success classifier predictions with a binary cross entropy loss between the predicted and ground truth success likelihood. We train for 500k iterations with batch size 64 on a V100 GPU which takes 5 days. We augment the data by rotating the combined object-scene point cloud by random 3D rotations to increase dataset diversity.

## A5 Experimental Setup

This section describes the details of our experimental setup in simulation and the real world.

### A5.1 Simulated Experimental Setup

We use PyBullet [17] and the AIRobot [73] library to set up the tasks in the simulation and quantitatively evaluate our method along with the baselines. The environment consists of a table with the shapes that make up the object and the scene, and the multiple simulated cameras that are used to obtain the fused 3D point cloud. We obtain segmentation masks of the object and the scene using PyBullet's built-in segmentation abilities.

### A5.2 Real World Experimental Setup

In the real world, we use a Franka Robot arm with a Robotiq 2F140 parallel jaw gripper attached for executing the predicted rearrangements. We also use four Realsense D415 RGB-D cameras with known extrinsic parameters. Two of these cameras are mounted to provide a clear, close-up view of the object, and the other two are positioned to provide a view of the scene objects. We use a combination of Mask-RCNN, density-based Euclidean clustering [74], and manual keypoint annotation to segment the object, and use simple cropping heuristics to segment the overall scene from the rest of the background/observation (e.g., remove the table and the robot from the observation so we just see the bookshelf with the books on it).

## A6 Evaluation Details

This section presents further details on the tasks we used in our experiments, the baseline methods we compared RPDiff against, and the mechanisms we used to apply the predicted rearrangement to the object in simulation and the real world.

### A6.1 Tasks and Evaluation Criteria

**Task Descriptions**. We consider three relational rearrangement tasks for evaluation: (1) hanging a mug on the hook of a rack, where there might be multiple racks on the table, and each rack might have multiple hooks, (2) inserting a book into one of the multiple open slots on a randomly posed bookshelf that is partially filled with existing books, and (3) placing a cylindrical can upright either on an existing stack of cans or on a flat open region of a shelf where there are no cans there. Each of these tasks features many placing solutions that achieve the desired relationship between the object and the scene (e.g., multiple slots and multiple orientations can be used for placing, multiple racks/hooks and multiple orientations about the hook can be used for hanging, multiple stacks and/or multiple regions in the cabinet can be used for placing the can, which itself can be placed with either flat side down and with any orientation about its cylindrical axis).

**Evaluation Metrics and Success Criteria**. To quantify performance, we report the average success rate over 100 trials, where we use the ground truth simulator state to compute success. For a trial to be successful, the object **O** and **S** must be in contact and the object must have the correct orientation relative to the scene (for instance, the books must be *on* the shelf, and must not be oriented with the long side facing into the shelf). For the can/cabinet task, we also ensure that the object **O** did not run into any existing stacks in the cabinet, to simulate the requirement of avoiding hitting the stacks and knocking them over.

We also quantify coverage via recall between the full set of predicted solutions and the precomputed set of solutions that are available for a given task instance. This is computed by finding the closest prediction to each of the precomputed solutions and checking whether the translation and rotation error between the prediction and the solution is within a threshold (we use 3.5cm for the translation and 5 degrees for the rotation). If the error is within this threshold, we count the solution as "detected". We compute recall for a trial as the total number of "detected solutions" divided by the total number of solutions available and report overall recall as the average over the 100 trials. Precision is computed in an analogous fashion but instead checks whether each prediction is within the threshold for at least one of the ground truth available solutions.

## A6.2 Baseline Implementation and Discussion

In this section, we elaborate on the implementation of each baseline approach in more detail and include further discussion on the observed behavior and failure modes of each approach.

### A6.2.1 Coarse-to-Fine Q-attention (C2F-QA).

C2F-QA adapts the classification-based approach proposed in [8], originally designed for pick-and-place with a fixed robotic gripper, to the problem of relational object rearrangement. We voxelize the scene and use a local PointNet [75] that operates on the points in each voxel to compute per-voxel input features. We then pass this voxel feature grid through a set of 3D convolution layers to compute an output voxel feature grid. Finally, the per-voxel output features are each passed through a shared MLP which predicts per-voxel scores. These scores are normalized with a softmax across the grid to represent a distribution of "action values" representing the "quality" of moving the centroid of the object to the center of each respective voxel. This architecture is based on the convolutional point cloud encoder used in Convolutional Occupany Networks [7].

To run in a coarse-to-fine fashion, we take the top-scoring voxel position (or the top-$k$ voxels if making multiple predictions), translate the object point cloud to this position, and crop the scene point cloud to a box around the object centroid position. From this cropped scene and the translated object, we form a combined object-scene input point cloud and re-voxelize just this local portion of the point cloud at a higher resolution. We then compute a new set of voxel features with a separate high-resolution convolutional point cloud encoder. Finally, we pool the output voxel features from this step and predict a distribution over a discrete set of rotations to apply to the object. We found difficulty in using the discretized Euler angle method that was applied in [8], and instead directly classify in a binned version of SO(3) by using an approximate uniform rotation discretization method that was used in [76].

We train the model to minimize the cross entropy loss for both the translation and the rotation (i.e., between the ground truth voxel coordinate containing the object centroid in the demonstrations and the ground truth discrete rotation bin). We use the same object point cloud perturbation scheme to create initial "noised" point clouds for the model to de-noise but have the model directly predict how to invert the perturbation transform in one step.

**Output coverage evaluation**. Since C2F-QA performs the best in terms of task success among all the baselines and is naturally suited for handling multi-modality by selecting more than just the `argmax` among the binned output solutions, we evaluate the ability of our method and C2F-QA to achieve high coverage among the available placing solutions while still achieving good precision (see Section 5.2). To obtain multiple output predictions from C2F-QA, we first select multiple voxel positions using the `top-k` voxel scores output by the PointNet → 3D CNN → MLP pipeline. We then copy the object point cloud and translate it to each of the selected voxel positions. For each selected position, we pool the local combined object-scene point cloud features and use the pooled features to predict a distribution of scores over the discrete space of rotations. Similar to selecting multiple

voxel positions, we select the `top-k` scoring rotations and use this full set of multiple translations + multiple rotations-per-translation as the set of output transforms to use for computing coverage.

**Relationship to other "discretize-then-classify" methods**. C2F-QA computes per-voxel features from the scene and uses these to output a normalized distribution of scores representing the quality of a "translation" action executed at each voxel coordinate. This idea of discretizing the scene and using each discrete location as a representation of a translational action has been successfully applied by a number of works in both 2D and 3D [41, 44, 77]. In most of these pipelines, the translations typically represent gripper positions, i.e., for grasping. In our case, the voxel coordinates represent a location to move the object for rearrangement.

However, techniques used by "discreteize-then-classify" methods for rotation prediction somewhat diverge. C2F-QA and the recently proposed PerceiverActor [44] directly classify the best discrete rotation based on pooled network features. On the other hand, TransporterNets [41] and O2O-Afford [43] exhaustively evaluate the quality of different rotation actions by "convolving" some representation of the object being rearranged (e.g., a local image patch or a segmented object point cloud) in *all* possible object orientations, with respect to *each* position in the entire discretized scene (e.g., each pixel in the overall image or each point in the full scene point cloud). The benefit is the ability to help the model more explicitly consider the "interaction affordance" between the object and the scene at various locations and object orientations and potentially make a more accurate prediction of the quality of each candidate rearrangement action. However, the downside of this "exhaustive search" approach is the computational and memory requirements are much greater, hence these methods have remained limited to lower dimensions.

### A6.2.2 Relational Neural Descriptor Fields (R-NDF).

R-NDF [18] uses a neural field shape representation trained on category-level 3D models of the objects used in the task. This consists of a PointNet encoder with SO(3)-equivariant Vector Neuron [78] layers and an MLP decoder. The decoder takes as input a 3D query point and the output of the point cloud encoder, and predicts either the occupancy or signed distance of the 3D query point relative to the shape. After training, a point or a rigid set of points in the vicinity of the shape can be encoded by recording their feature activations of the MLP decoder. The corresponding point/point set relative to a new shape can then be found by locating the point/point set with the most similar decoder activations. These point sets can be used to parameterize the pose of local oriented coordinate frames, which can represent the pose of a secondary object or a gripper that must interact with the encoded object.

R-NDFs have been used to perform relational rearrangement tasks via the process of encoding task-relevant coordinate frames near the object parts that must align to achieve the desired rearrangement, and then localizing the corresponding parts on test-time objects so a relative transform that aligns them can be computed. We use the point clouds from the demonstrations to record a set of task-relevant coordinate frames that must be localized at test time to perform each of the tasks in our experiments. The main downside of R-NDF is if the neural field representation fails to faithfully represent the shape category, the downstream corresponding matching also tends to fail. Indeed, owing to the global point cloud encoding used by R-NDF, the reconstruction quality on our multi-rack/bookshelf/cabinet scenes is quite poor, so the subsequent correspondence matching does not perform well on any of the tasks we consider.

### A6.2.3 Neural Shape Mating (NSM) + CVAE.

Neural Shape Mating (NSM) [3] uses a Transformer to process a pair of point clouds and predict how to align them. The method was originally deployed on the task of "mating" two parts of an object that has been broken but can be easily repurposed for the analogous task of relational rearrangement given a point cloud of a manipulated object and a point cloud of a scene/"parent object". Architecturally, NSM is the same as our relative pose regression model, with the key differences of (i) being trained on arbitrarily large perturbations of the demonstration point clouds, (ii) not using local cropping, and (iii) only making a single prediction. We call this baseline "NSM-base" because we do not consider the auxiliary signed-distance prediction and learned discriminator proposed in the original approach [3]. As shown in Table 1, the standard version of NSM fails to perform well on any of the tasks that feature multi-modality in the solution space (nor can the model successfully fit the demonstration data). Therefore, we adapted it into a conditional variational autoencoder (CVAE) that at least has the capacity to learn from multi-modal data and output a distribution of transformations.

We use the same Transformer architecture for both the CVAE encoder and decoder with some small modifications to the inputs and outputs to accommodate (i) the encoder also encoding the ground truth de-noising transforms and predicting a latent variable $z$, and (ii) the decoder conditioning on $z$ in addition to the combined object-scene point cloud to reconstruct the transform. We implement this with the same method that was used to incorporate the timestep information in our architecture – for the encoder, we include the ground truth transform as both an additional input token and via a residual connection with the global output feature, and for the decoder, we include the latent variable in the same fashion. We also experimented with concatenating the residually connected features and did not find any benefit. We experimented with different latent variable dimensions and weighting coefficients for the reconstruction and the KL divergence loss terms, since the CVAE models still struggled to fit the data well when the KL loss weight was too high relative to the reconstruction. However, despite this tuning to enable the CVAE to fit the training data well, we found it struggled to perform well at test time on unseen objects and scenes.

### A6.3 Common failure modes

This section discusses some of the common failure modes for each method on our three tasks.

For **Book/Shelf**, our method occasionally outputs a solution that ignores an existing book already placed in the shelf. We also sometimes face slight imprecision in either the translation or rotation prevents the book from being able to be inserted. Similarly, the main failure modes on this task from the baselines are more severe imprecision. C2F-QA is very good at predicting voxel positions accurately (i.e., detecting voxels near open slots of the shelf) and the rotation predictions are regularly close to something that would work for book placement, but the predicted book orientations are regularly too misaligned with the shelf to allow the insertion to be completed.

For **Mug/Rack**, a scenario where our predictions sometimes fail is when there is a tight fit between the nearby peg and the handle of the mug. For C2F-QA, the predictions appear to regularly ignore the location of the handle when orienting the mug – the positions are typically reasonable (e.g., right next to one of the pegs on a rack) but the orientation oftentimes appears arbitrary. We also find C2F-QA achieves the highest training loss on this task (and hypothesize this occurs for the same reason).

Finally, for **Can/Cabinet**, a common failure mode across the board is predicting a can position that causes a collision between the can being placed and an existing stack of cans, which we don't allow to simulate the requirement of avoiding knocking over an existing stack.

### A6.4 Task Execution

This section describes additional details about the pipelines used for executing the inferred relations in simulation and the real world.

#### A6.4.1 Simulated Execution Pipeline

The evaluation pipeline mirrors the demonstration setup. Objects from the 3D model dataset for the respective categories are loaded into the scene with randomly sampled position and orientation. We sample a rotation matrix uniformly from $SO(3)$, load the object with this orientation, and constrain the object in the world frame to be fixed in this orientation. We do not allow it to fall on the table under gravity, as this would bias the distribution of orientations covered to be those that are stable on a horizontal surface, whereas we want to evaluate the ability of each method to generalize over all of $SO(3)$. In both cases, we randomly sample a position on/above the table that are in view for the simulated cameras.

After loading object and the scene, we obtain point clouds $\mathbf{P_O}$ and $\mathbf{P_S}$ and use RPDiff to obtain a rearrangement transform to execute. The predicted transformation is applied by resetting the object state to a "pre-placement" pose and directly actuating the object with a position controller to follow a straight-line path. Task success is then checked based on the criteria described in the section above.

**Pre-placement Offset and Insertion Controller**. Complications with automatic success evaluation can arise when directly resetting the object state based on the predicted transform. To avoid such complications, we simulate a process that mimics a closed-loop controller executing the last few inches of the predicted rearrangement from a "pre-placement" pose that is a pure translational offset from the final predicted placement. For our quantitative evaluations, we use the ground truth state of

the objects in the simulator together with prior knowledge about the task to determine the direction of this translational offset. For the mug/rack task, we determine the axis that goes through the handle and offset by a fixed distance in the direction of this axis (taking care to ensure it does not go in the opposite direction that would cause an approach from the wrong side of the rack). For the can/cabinet task and the book/bookshelf task, we use the known top-down yaw component of the shelf/cabinet world frame orientation to obtain a direction that offsets along the opening of the shelf/cabinet.

To execute the final insertion, we reset to the computed pre-placement pose and directly actuate the object with a position controller to follow a straight line path from the pre-placement pose to the final predicted placement. To simulate some amount of reactivity that such an insertion controller would likely possess in a full-stack rearrangement system, we use the simulator to query contact forces that are detected between the object and the scene. If the object pose is not close to the final predicted pose when contacts are detected, we back off and sample a small "delta" translation and body-frame rotation to apply to the object before attempting another straight-line insertion. These small adjustments are attempted up to a maximum of 10 times before the execution is counted as a failure. If, upon detecting contact between the object and the scene, the object is within a threshold of its predicted place pose, the controller is stopped and the object is dropped and allowed to fall under gravity (which either allows it to settle stably in its final placement among the scene object, or causes it to fall away from the scene). We use this same procedure across all methods that we evaluated in our experiments.

We use this combination of a heuristically-computed pre-placement pose and "trial-and-error" insertion controller because (i) it removes the need for a full object-path planning component that searches for a feasible path the object should follow to the predicted placement pose (as this planning problem would be very challenging to solve to due all the nearby collisions between the object and the scene), (ii) it helps avoid other artificial execution failures that can arise when we perform the insertion from the pre-placement pose in a purely open-loop fashion, and (iii) it enables us to avoid complications that can arise from directly resetting the object state based on the predicted rearrangement transform. However, we also observe some failure modes and brittleness that arises from our use of manual computation and heuristics to compute these pre-placement poses, and in the future, we would like to explore predicting additional feasible waypoint poses that help construct a full path from start to goal for the object. Below, we include further details and discussion on the heuristics used for computing the pre-placement offsets in simulation.

### A6.4.2   Computing pre-placement offset poses with task-specific heuristics in simulation

Future versions ought to introduce predictions of more intermediate waypoints (note diffusion has shown to be useful in this context as well, e.g,. for motion planning/trajectory modeling [1, 2, 3]).

**Book/bookshelf and Can/cabinet**. Since the simulator pose of each object is available, we use the top-down orientation of the shelf/cabinet to obtain the offset vector. The $[x, y]$ world-frame components of the vector are computed such that, from a top-down perspective, the 2D vector is perpendicular to the front opening of the shelf/cabinet. The $z$ component of the vector is set to 0. This allows the books/cans to be moved to the vicinity of the final placement, with a pure 2D offset such that moving along this offset in a straight line can achieve successful insertion/stacking. If the orientation or the position of the predicted pose is wrong following the 2D vector from the offset version of these poses can cause the insertion/placement to fail. Example reasons for this failure include the book not fitting, due to an incorrect orientation, and the can colliding with one of the existing stacks (which we check for and count as a failure).

**Mug/rack**. We use simulated ShapeNet mugs that have a canonical orientation. Based on this canonical orientation, we know the 3D vector direction corresponding to a vector that points through the opening of the handle on the mug. This vector can point in two different directions; we select the one with the larger $+z$ component, based on the knowledge that the mug should approach the rack from above (since the hooks are angled slightly upward, to avoid the mugs falling when they are hung). Using this vector, we translate the mug from its predicted hanging pose along a direction that goes through the handle, so that when we actuate it from this offset (assuming the prediction is accurate), the hook ends up going through the handle. If the position or orientation of the prediction is incorrect, then the offset pose will be computed so that the mug either cannot be placed on the rack (due to collisions occurring between the handle and the hook) or the placement will miss the hook entirely (so the mug falls away and fails to be hung) – both of these cases are treated as failures.

### A6.4.3 Real World Execution Pipeline

Here, we repeat the description of how we execute the inferred transformation using a robot arm with additional details. At test time, we are given point clouds $\mathbf{P_O}$ and $\mathbf{P_S}$ of object and scene, and we obtain $\mathbf{T}$, the SE(3) transform to apply to the object from RPDiff. $\mathbf{T}$ is applied to $\mathbf{O}$ by transforming an initial grasp pose $\mathbf{T}_{\text{grasp}}$, which is obtained using a separate grasp predictor [10], by $\mathbf{T}$ to obtain a placing pose $\mathbf{T}_{\text{place}} = \mathbf{T}\mathbf{T}_{\text{grasp}}$. As in the simulation setup, we use a set of task-dependent heuristics to compute an additional "pre-placement" pose $\mathbf{T}_{\text{pre-place}}$, from which we follow a straight-line end-effector path to reach $\mathbf{T}_{\text{place}}$. We then use off-the-shelf inverse kinematics and motion planning to move the end-effector to $\mathbf{T}_{\text{grasp}}$ and $\mathbf{T}_{\text{place}}$.

To ease the burden of collision-free planning with a grasped object whose 3D geometry is unknown, we also compute an additional set of pre-grasp and post-grasp waypoints which are likely to avoid causing collisions between the gripper and the object during the execution to the grasp pose, and collisions between the object and the table or the rest of the scene when moving the object to the pre-placement pose. Each phase of the overall path is executed by following the joint trajectory in position control mode and opening/closing the fingers at the correct respective steps. The whole pipeline can be run multiple times in case the planner returns infeasibility, as the inference methods for both grasp and placement generation have the capacity to produce multiple solutions.

### A6.4.4 Computing pre-placement offset poses with task-specific heuristics in the real world

Here, we describe details on the heuristics we used to compute the pre-placement poses are included in the subsection below. We acknowledge that noise in the computation of these pre-placement poses was a common source of execution failure in our real-world qualitative trials, and future work that also learns to robustly predict additional feasible waypoint poses that help reach the final placement is likely to support improved rearrangement performance in the real world.

**Book/bookshelf**. In the real world, we again use the knowledge that the placement offset should primarily consist of a 2D $[x, y]$ translation from the predicted pose on the shelf. We fit an oriented bounding box to the predicted book point cloud and select the 2D vector corresponding to the longest corner on the bottom face of the bounding box (with a small $+z$ component, to help the placement avoid clipping the shelf with the bottom of the book by approaching from slightly above). Again, this 2D vector can have two potential directions, and we select the one that points toward the center of the table (assuming we are not placing on a shelf from the far edges of the table).

**Can/cabinet**. For the real-world can stacking task, we cropped a portion of the cabinet point cloud (to avoid any outlier points from the table), fit a bounding box to it, and selected the 2D vector corresponding to the corner on the bottom face of the bounding box that pointed from the center of the cabinet point cloud most closely to the center of the table (again, assuming we were approaching from near the center of the table, rather than the table edges).

**Mug/rack**. In the real world, we attempt to approximate the 3D offset vector based on fitting a 3D line to the part of the point cloud corresponding to the nearby hook. Due to noise in the point cloud and an imperfect ability to solely segment out the hook from the body of the rack, this offset computation was the least robust and introduced some failed execution attempts.

## A7  Extra Ablations

In this section, we perform additional experiments wherein different system components are modified and/or ablated.

**With vs. Without Success Classifier**. We use neural network $h_\phi$ to act as a success classifier and support selecting a "best" output among the $K$ predictions made by our iterative de-noising procedure. Another simple mechanism for selecting an output index $k_{\text{exec}}$ for execution would be to uniformly sample among the $K$ outputs. However, due to the local nature of the predictions at small values of $t$ and the random guess initializations used to begin the inference procedure, some final solutions end in configurations that don't satisfy the task (see the book poses that converge to a region where there is no available slot for placement in Figure A5 for $A = 10$).

Therefore, a secondary benefit of incorporating $h_\phi$ is to filter out predictions that may have converged to these "locally optimal" solutions, as these resemble some of the negatives that the classifier has

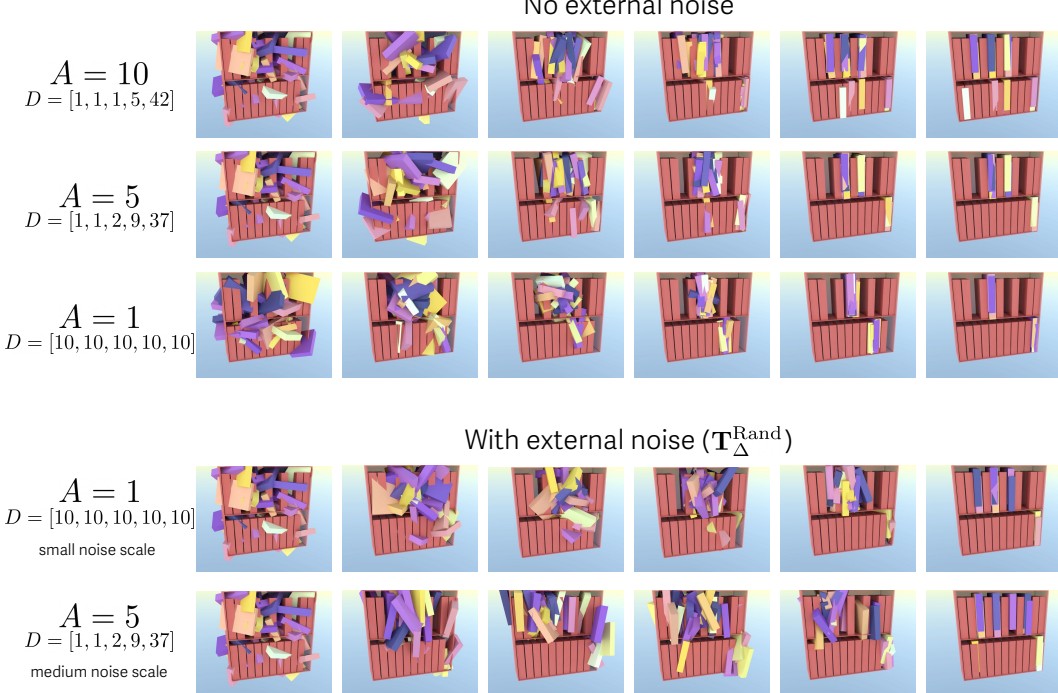

Figure A6: Examples of running our full iterative evaluation procedure for $I$ steps with different values of $A$ (and subsequently, $D$) in our `i_to_t` function (which maps from test-time iteration values $n = 1, ..., I$ to the timestep values $t = 1, .., T$ that were used in training), and with different amounts of external noise $\mathbf{T}_\Delta^{\mathrm{Rand}}$ added from the annealed external noise distribution $p_{\mathrm{AnnealedRandSE(3)}}(\cdot)$. We observe that with large values of $A$, the model makes more predictions with smaller values of $t$. These predictions are more local and the overall solutions converge to a more broad set of rearrangement transforms. This sometimes leads to "locally optimal" solutions that fail at the desired task (see top right corner with $A = 10$). With small $A$, the early iterations are more biased toward the average of a general region, so the set of transforms tends to collapse on a single solution within a region. By incorporating external noise, a better balance of coverage for smaller values of $A$ and "local optima" avoidance for larger values of $A$ can be obtained.

seen during training. Indeed, we find the average success rate across tasks with RPDiff when using the success classifier is 0.88, while the average success when uniformly sampling the output predictions is 0.83. This difference is relatively marginal, indicating that the majority of the predictions made by the pose de-noising procedure in RPDiff are precise enough to achieve the task. However, the performance gap indicates that there is an additional benefit of using a final success classifier to rank and filter the outputs based on predicted success.

**Noise vs. No Noise**. In each update of the iterative evaluation procedure, we update the overall predicted pose and the object point cloud by a combination of a transform predicted by $f_\theta$ and a randomly sampled "external noise" transform $\mathbf{T}_\Delta^{\mathrm{Rand}}$. The distribution that $\mathbf{T}_\Delta^{\mathrm{Rand}}$ is sampled from is parameterized by the iteration number $i$ to converge toward producing an identity transform so the final pose updates are purely a function of the network $f_\theta$.

The benefit of incorporating the external noise is to better balance between precision and coverage. First, external noise helps the pose/point cloud at each iteration "bounce out" of any locally optimal regions and end up near regions where a high quality solution exists. Furthermore, if there are many high-quality solutions close together, the external noise on later iterations helps maintain some variation in the pose so that more overall diversity is obtained in the final set of transform solutions.

For instance, see the qualitative comparisons in Figure A6 that include iterative predictions both with and without external noise. For a value of $A = 1$ in `i_to_t`, only two of the available shelf slots are found when no noise is included. With noise, however, the method finds placements that cover four of the available slots. Quantitatively, we also find that incorporating external noise helps in terms of

overall success rate and coverage achieved across tasks. The average $($Success Rate, Recall$)$ across our three tasks with and without noise was found to be (0.88, 0.44) and (0.83, 0.36), respectively.

**Number of diffusion steps $T$ during training**. The total number of steps $T$ and the noise distribution for obtaining perturbation a transform $\mathbf{T}_{\text{noise}}^{(t)}$ affects the magnitude of the translation and rotation predictions that must be made by the model $f_\theta$. While we did not exhaustively search over these hyperparameters, early in our experiments we found that very small values of $T$ (e.g., $T = 2$) cause the predictions to be much more imprecise. This is due to the averaging that occurs between training samples when they are too far away from the ground truth. In this regime, the examples almost always "look multi-modal" to the model. On the other hand, for large values of $T$ (e.g., $T = 50$), the incremental transforms that are used to de-noise become very small and close to the identity transform. When deployed, models trained on this data end up failing to move the object from its initial configuration because the network has only learned to make extremely small pose updates.

We found a moderate value of $T = 5$ works well across each of our tasks, though other similar values in this range can likely also provide good performance. This approximately leads the average output scale of the model to be near 2.5cm translation and 18-degree rotation. We also observe a benefit in biasing sampling for the timesteps $t = 1, ..., T$ to focus on smaller values. This causes the model to see more examples close to the ground truth and make more precise predictions on later iterations during deployment. We achieve this biased sampling by sampling $t$ from an exponentially decaying categorical probability distribution over discrete values $1, 2, ..., T$.

**Incremental targets vs. full targets**. As discussed in Section 3.1, encouraging the network $f_\theta$ to predict values with roughly equal magnitude is beneficial. To confirm this observation from the literature, we quantitatively evaluate a version of the de-noising model $f_\theta$ trained to predict the full de-noising transform $\left[\mathbf{T}_{\text{noise}}^{(t)}\right]^{-1}$. The quantitative $($Success Rate, Recall$)$ results averaged across our three tasks with the incremental de-noising targets are $(0.88, 0.44)$, while the model trained on full de-noising targets are $(0.76, 0.34)$. These results indicate a net benefit in using the incremental transforms as de-noising prediction targets during training.

**Value of $A$ in i_to_t**. In this section, we discuss the effect of the value $A$ in the i_to_t function used during the iterative evaluation procedure. The function i_to_t maps evaluation iteration values $i$ to timestep values $t$ that were seen during training. For instance, we may run the evaluation procedure for 50 iterations, while the model may have only been trained to take values up to $t = 5$ as input. Our i_to_t function is parameterized by $A$ such that larger values of $A$ lead to more evaluation iterations with small values of $t$. As $A$ approaches 1, the number of iterations for each value of $t$ becomes equal (i.e., for $A = 1$, the number of predictions made for each value of $t$ is equal to $I/T$).

Figure A6 shows qualitative visualizations of de-noising the pose of a book relative to a shelf with multiple available slots with different values of $A$ in the i_to_t function. This example shows that the solutions are more biased to converge toward a single solution for smaller values of $A$. This is because more of the predictions use larger values of $t$, which correspond to perturbed point clouds that are farther from the ground truth in training. For these perturbed point clouds, their association with the correct target pose compared to other nearby placement regions is more ambiguous. Thus, for large $t$, the model learns an averaged-out solution that is biased toward a region near the average of multiple placement regions that may be close together. On the other hand, for large $A$, more predictions correspond to small values of $t$ like $t = 1$ and $t = 0$. For these timesteps, the model has learned to precisely snap onto whatever solutions may exist nearby. Hence, the pose updates are more local and the overall coverage across the $K$ parallel runs is higher. The tradeoff is that these predictions are more likely to remain stuck near a "locally optimal" region where a valid placement pose may not exist. Table 3 shows the quantitative performance variation on the **Book/Shelf** task for different values of $A$ in the i_to_t function. These results reflect the trend toward higher coverage and marginally lower success rate for larger values of $A$.

## A8 Further Discussion on Real-world System Engineering and Limitations

This section provides more details on executing rearrangement via pick-and-place on the real robot (to obtain the results shown in Figures 1 and 4) and discusses additional limitations of our approach.

| Metric | Value of $A$ in `i_to_t` | | | | |
|---|---|---|---|---|---|
| | 1 | 2 | 5 | 10 | 20 |
| **Success Rate** | 1.00 | 0.95 | 0.96 | 0.94 | 0.90 |
| **Recall (coverage)** | 0.37 | 0.41 | 0.48 | 0.48 | 0.52 |

Table 3: Performance for different values of $A$ in `i_to_t`. Larger values of $A$ obtain marginally better precision at the expense of worse coverage (lower recall).

### A8.1    Executing multiple predicted transforms in sequence in real-world experiments

The output of the pose diffusion process in RPDiff is a set of $K$ SE(3) transforms $\{\mathbf{T}_k^{(0)}\}_{k=1}^K$. To select one for execution, we typically score the outputs with success classifier $h_\phi$ and search through the solutions while considering other feasibility constraints such as collision avoidance and robot workspace limits. However, to showcase executing a diverse set of solutions in our real-world experiments, a human operator performs a final step of visually inspecting the set of feasible solutions and deciding which one to execute. This was mainly performed to ease the burden of recording robot executions that span the space of different solutions (i.e., to avoid the robot executing multiple similar solutions, which would fail to showcase the diversity of the solutions produced by our method).

### A8.2    Expanded set of limitations and limiting assumptions

Section 7 mentions some of the key limitations of our approach. Here, we elaborate on these and discuss additional limitations, as well as potential avenues for resolving them in future work.

- We train from scratch on demonstrations and do not leverage any pre-training or feature-sharing across multiple tasks. This means we require many demonstrations for training. A consequence of this is that we cannot easily provide enough demonstrations to train the diffusion model in the real world (while still enabling it to generalize to unseen shapes, poses, and scene layouts). Furthermore, because we train only in simulation and directly transfer to the real world, the domain gap causes some challenges in sim2real transfer, so we do observe worse overall prediction performance in the real world. This could be mitigated if the number of demonstrations required was lower and we could train the model directly on point clouds that appear similar to those seen during deployment.

- In both simulation and the real world, we manually completed offset poses for moving the object before executing the final placement. A more ideal prediction pipeline would involve generating "waypoint poses" along the path to the desired placement (or even the full collision-free path, e.g., as in [79]) to support the full insertion trajectory rather than just specifying the final pose.

- Our method operates using a purely geometric representation of the object and scene. As such, there is no notion of physical/contact interaction between the object and the scene. If physical interactions were considered in addition to purely geometric/kinematic interactions/alignment, the method may be even more capable of accurate final placement prediction and avoid some of the small errors that sometimes occur. For instance, a common error in hanging a mug on a rack is to have the handle *just* miss the hook on the rack. While these failed solutions are geometrically very close to being correct, physically, they are completely different (i.e., in one, contact occurs between the two shapes, while in the other, there is no contact that can support the mug hanging).

- Our method operates using 3D point clouds which are currently obtained from depth cameras. While this permits us to perform rearrangements with a wide variety of real-world objects/scenes that can be sensed by depth cameras, there are many objects which cannot be observed by depth cameras (e.g., thin, shiny, transparent objects). Investigating a way to perform similar relational object-scene reasoning in 6D using signals extracted from RGB sensors would be an exciting avenue to investigate.

## A9    Additional Results

### A9.1    Training multi-task models for pose de-noising

While Section 5 shows results for models trained on datasets corresponding to single tasks, here, we discuss preliminary results on training one model on data from all the tasks together. In particular, we trained the diffusion model $f_\theta$ on the combined set of demonstrations across all three tasks and evaluated its performance on held-out test instances of each task. The average success rate across tasks was 85%, which is comparable to the performance achieved by the single-task models (88%).

### A9.2    Multi-step Manipulation with RPDiff

In this section, we provide a qualitative example of how RPDiff can be used to support predicting and executing rearrangement actions for tasks requiring multiple steps and/or manipulating multiple objects in sequence. We use the example of placing three books on a table into a shelf, one by one.

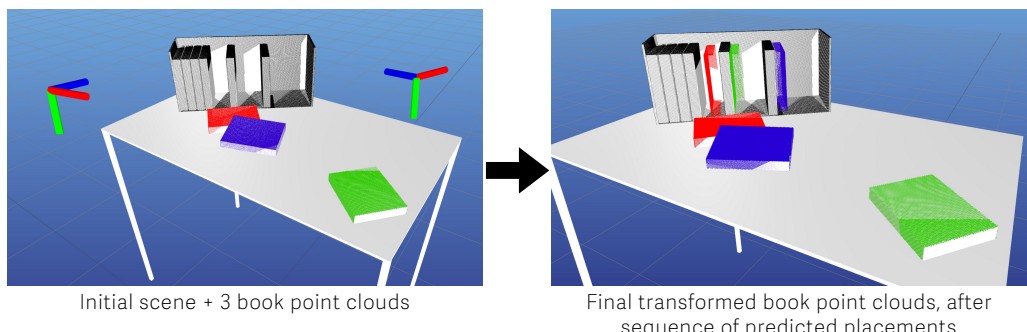

Initial scene + 3 book point clouds        Final transformed book point clouds, after
                                            sequence of predicted placements

Figure A7: **(Left)** Initial environment with three books on a table (with corresponding point clouds shown in green, red, and blue) along with a bookshelf with multiple open slots (bookshelf point cloud shown in black). The task is to place *all three books* into the shelf by sequentially predicting transforms that should be applied to each of them. We will use RPDiff to achieve this by cycling through each book point cloud and updating the corresponding scene point cloud on each step. **(Right)** The predicted transform for each book is shown (see the green, red, and blue point clouds transformed into configurations on the shelf).

In Fig. A7 (left), we show the original scene with three books and a shelf. In Fig. A7 (right), we show the same scene and initial objects, along with corresponding *transformed* objects (with colors indicating which initial and final book point clouds go together). These final point clouds have been obtained by *sequentially* (i) inferring a relative SE(3) transform of one of the books, followed by (ii) modifying the scene point cloud to include the newly-transformed book point cloud, so that it can be considered as part of the scene on the prediction of where to place the next book.

To begin, the first book (picked at random - shown in blue in Fig. A8) is transformed into the configuration shown in yellow (Fig. A8, left) using a prediction from RPDiff. Subsequently, the transformed book point cloud is added to the point cloud representing the scene (Fig. A8, right). Next, we perform the same process (i.e. apply RPDiff with a new book point cloud, shown in blue in Fig. A9) with the *new* scene point cloud resulting from step one. Note this could be "imagined" by directly applying the predicted transform to the originally-observed point cloud of the first book. Alternatively, we could *execute* the first predicted step and then re-perceive the whole scene (which would now include the just-placed first book). For simplicity, we have shown "imagining" the new scene point cloud by transforming the originally-observed book point cloud based on the predicted placement transform. Finally, we repeat this process for a third step with the remaining book (shown in blue in Fig. A10), again updating the "full" scene point cloud to reflect the placement of the first two books (see Fig. A9, right). Note that each of these placements was selected as the maximum-scoring prediction as evaluated by our success classifier.

Overall, as shown in Fig. A7 (right), we have obtained a set of SE(3) transforms corresponding to each of the three books on the table, and by either imagining the execution of each step or performing the execution of each step (and then re-perceiving), we can take into account the placement of earlier books when having RPDiff infer where to place the next books.

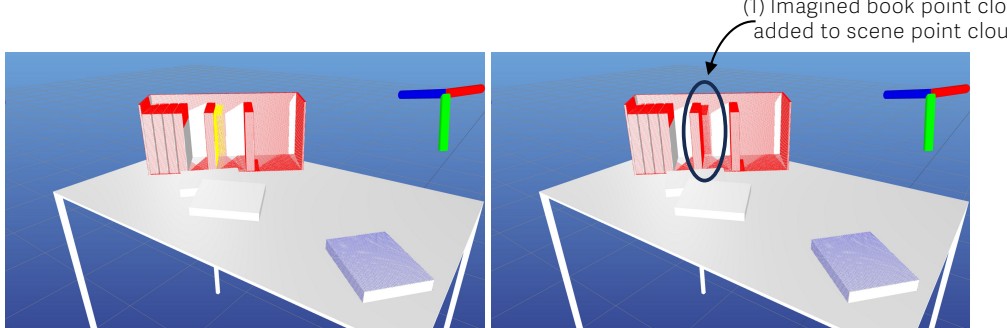

Figure A8: Multi-step book placement, step 1. The first book is selected at random (point cloud shown in blue) and RPDiff is used to obtain a transform for placing this book on the shelf. The transformed point cloud is shown in yellow on the **left**, and the corresponding *new* scene point cloud (with the transformed book point cloud included) is shown in red on the **right**.

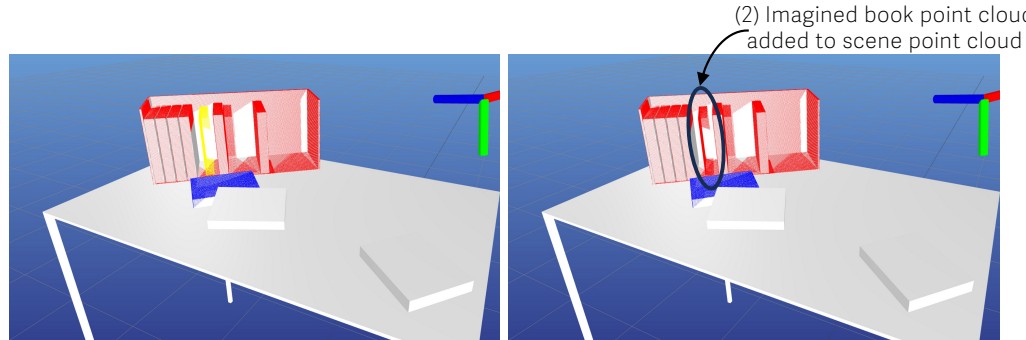

Figure A9: Multi-step book placement, step 2. The updated "full scene point cloud" from step 1 is used as the input scene point cloud to step 2. The second book is selected (point cloud shown in blue) and a corresponding transform is obtained with RPDiff. Once again, the transformed point cloud is shown in yellow on the **left**, and the corresponding *new* scene point cloud (with the transformed book point cloud included) is shown in red on the **right**.

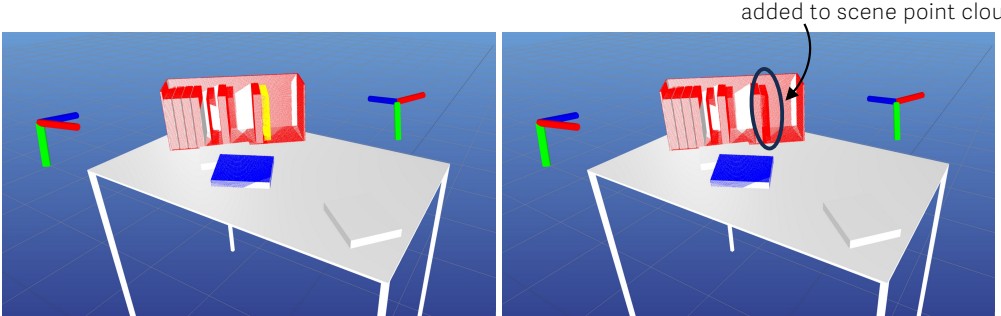

Figure A10: Multi-step book placement, step 3. This process is repeated a final time with the third book (point cloud shown in blue) and the scene point cloud with both other books in their "imagined" placement configuration. RPDiff predicts a rearrangement transform to place the third book (shown in yellow on the **left**), such that the process could be continued with the newly updated scene point cloud (**right**) if needed.

## A10 Expanded Discussion

### A10.1 Demo collection in simulation with manually-designed heuristics, scripted policies, and known 3D models

Here we clarify the details and context for our specific choice of data generation/demonstration collection procedure and mention alternatives that could have been employed instead. While the specific approach taken in this work involves scripted mechanisms for generating demonstrations of each rearrangement task, it is important to note our method does *not* fundamentally rely on these components. We only rely on demonstrations of object-scene point clouds in final configurations that satisfy the desired object-scene relation. These demonstrations could equivalently come from the real world (i.e., via teleoperation or kinesthetic teaching, similar to [18, 37]), but we were unable to collect a large number of these demonstrations in the real world, and instead opted to automatically collect them in simulation. With this goal, we take advantage of privileged simulator and model information, as well as task-specific heuristics, purely for automating data generation. This information includes 3D models, simulator states, canonical object poses, and available placing locations in the scenes (which were recorded when scene objects were generated). One goal of future work is to bring down the number of demos required so the training examples can be directly shown in the real world on real objects

Related to this choice of collecting demos with scripted policies and manually designed heuristics, potential questions and concerns may arise regarding why the same heuristics could not have been used to solve the tasks in the first place. To this point, we emphasize that the information used to generate the demos is *not* available in the real world, and that we only use the heuristics and privileged information to generate training data. From this data, we obtain models that operate with point clouds, and these models *can* be deployed in the real world on unknown objects and scenes due to their use of a representation that can be obtained from standard perception systems. This is somewhat analogous to the paradigm of teacher-student training in reinforcement learning and imitation learning, wherein a "teacher" policy is trained using privileged information and then distilled to a "student" policy that operates from perception [80, 81].

### A10.2 Scripted demo collection using task-specific heuristics and privileged information

**Book/Shelf Placing**. We manually generated the bookshelf 3D models as shelves with books placed randomly among the available locations on each shelf. In creating these shelf models, we placed books in some of the existing slot locations and recorded locations and orientations near each of the remaining open slots. These open slot poses were then used when generating object-scene point cloud demonstrations, so that we could directly obtain SE(3) transforms that move books from their initial poses into final poses within the shelf. The real-world alternative would be to manually configure a diverse set of shelves and book placements within the shelves, and collect demonstrations of placing new books on these shelves.

**Can/Cabinet Stacking**. Similar to the approach for book/shelf placing, we programmatically generated the 3D scenes with cabinets and existing stacks of cans and recorded poses of available poses either on top of the existing can stacks or in regions large enough for making new stacks. Again, we use this information in the data generation phase to directly transform new cans into configurations that are either "on" an existing stack or begin a new stack in an open area. The real-world alternative would be to manually configure a diverse set of cabinets and can stacks within the cabinets. With these scenes, we could then collect demonstrations of placing/stacking new cans within these cabinets.

**Mug/Rack-Multi Hanging**. The mugs come from ShapeNet, where the 3D models for each category are all canonically aligned (i.e., the opening is aligned with the $+y$ axis and the handle is aligned with the $+z$ axis of the body frame). We use knowledge of these canonical poses to approximate the pose of the handle on each mug and similarly use approximate knowledge of the hook on each rack to roughly align the mug handle with the hook of a rack. Since these estimates are not perfect, we sample random perturbations of these poses until one is found that (1) does not cause collision between the objects and (2) leads to stable hanging of the mug on the rack when it falls under gravity. Similar to the above two tasks, with a few racks and mug instances, such demos could be collected in the real world without introducing any such assumptions.

### A10.3 Expanded Performance Analysis and Discussion

Here we provide further analysis and characterization of the system performance.

#### A10.3.1 Simulation performance breakdown

While our simulation success rates are high, there remains room for improved performance. First, we did not achieve complete coverage over all possible modes in the space of placing solutions (e.g., note that recall terminates near 0.68 in Fig. 3a). We find that there remains some bias toward and/or away from certain modes. This could possibly arise from bias or spurious correlations in the object point clouds and demonstrated placements. For example, the initial book point clouds often had one large visible face, with the other less visible (due to laying flat on the table). It is possible that the data contained a spurious correlation where the visible side was aligned with the open slot in a particular orientation more frequently, thus resulting in the model acquiring this bias. Another observation is that sometimes two modes are located very close together (e.g., two open slots might be directly next to each other). In these cases, the model may have again learned a biased solution either preferring one over the other or reaching an average between these close-together solutions. Either of these would lead to the less-visited/separate solutions to be predicted less frequently.

The model also occasionally tries to place objects in parts of the scene that cannot be reached (e.g., placing a book where there is already a book). This is sometimes due to ambiguity in the scene object poses (i.e., it can be hard to distinguish the front vs. back of the shelf). Finally, the model sometimes places an object "just off". For example, the handle of the mug is aligned to the rack, but is not far enough "on" the hook, resulting in the mug just missing the hook when dropped under gravity. Similarly, predicted can placements are sometimes very close but cause a collision with the stack of cans below. This highlights the large precision demands placed on a system that can perform rearrangement with high performance.

#### A10.3.2 Real world performance breakdown and sim2real challenges

**Placement prediction accuracy**. The distribution shift between simulated and real point clouds appeared to be the largest source of lower-quality transform predictions that did not succeed at placing. This shift is in part caused by depth sensor noise, point cloud outliers, imperfect camera calibration, and shiny object surfaces. Another source of distribution shift is generated training scenes not having perfect realism. Sim2real pipelines are still heavily bottlenecked by the effort of creating highly realistic 3D assets *and* configuring diverse yet structurally/physically valid scene layouts to train on.

We explored some techniques to mitigate the negative effects of sim2real gaps such as adding small per-point Gaussian noise to the point clouds and simulating additional occlusions from randomly posed synthetic cameras. We observed some marginal benefit in applying these techniques but did not rigorously evaluate how much they helped. Other ideas to explore for reducing sim2real gaps with 3D point clouds include using predicted depth models that can produce cleaner depth images than our depth sensors, using better depth sensors, and training and deploying more high-fidelity noise models to augment the simulation data and make it more similar to the real world.

**Robot execution success**. One aspect of robot execution success is unaccounted object motion that occurs while grasping/moving the object. We made the simplifying assumption that that object does not move during grasping and that, after grasping, the object is rigidly coupled to the gripper. If there is object movement during grasping/trajectory execution, the placement may become inaccurate. One way to mitigate this would be to estimate object motion post-grasp and account for this when reaching the placing pose. We did not implement this or similar system engineering improvements but it should be straightforward to pursue in the future.

Another source of failure was imperfect computation of task-specific pre-placement offset poses. Noise in the point cloud and brittleness of the heuristics we used sometimes meant the pre-placement offset we obtained did not allow a feasible approach to reach the predicted pose, even when our final pose predictions looked great. In future versions of the system, we want to incorporate predictions of additional feasible waypoints along the *path* to the placement.

Finally, another limitation having more to do with execution efficiency is the planning times required for searching for IK solutions and motion plans. Improvements in this respect are orthogonal to our

primary objective of predicting object placements that satisfy the desired relation but are certainly important for future versions of the system to operate more effectively.

### A10.4 Computational efficiency

We timed the forward pass (with our hyperparameters) to require 280ms with $K$=32 and 49ms with $K$=1. We also measured the time required for 50 eval iterations. It took 3.3s with $K$=1 and 24.46s with $K$=32 (all on a V100 GPU). Multi-step inference is mainly bottlenecked by other operations like (re-)cropping the scene point cloud - our batched implementations of these can be improved to reduce runtime. We leave optimizing computational speed for real-time performance to future work, but we're optimistic based on the results shown in [56] and our observations of RPDiff achieving good performance with as few as 10 iterations.

### A10.5 Additional Related Work Discussion

**OmniHang** [46] proposes a multi-stage approach for generating hanging placements in a category-agnostic fashion. Their pipeline involves coarse hanging pose regression, keypoint detection and alignment, and refinement via the cross-entropy method (CEM) with a learned reward function. Their first stage is analogous to our NSM + CVAE baseline, which we found to perform poorly due to the limits of predicting a full transform in a single step and constraining the distribution of latent variables to follow a unimodal Gaussian. The second stage of OmniHang is analogous to our R-NDF baseline, which localizes task-relevant object parts and brings them into alignment. We could have explored using a supervised keypoint detector, as in OmniHang, instead of matching features learned from self-supervision as in R-NDF, but this would have required additional manual keypoint labeling. Finally, stage three of OmniHang is analogous to using our learned success classifier to refine the prediction by performing local optimization, i.e., to maximize the score of the classifier (this has also been deployed in other work for 6-DoF grasp generation [82]). However, optimizing a learned binary classifier can be susceptible to finding solutions in out-of-distribution regions where the classifier outputs erroneously large scores, thus requiring extra components to constrain the search to a local region. Other related work has compared against similar baselines of optimizing learned cost functions [38, 54] and found it can be difficult to achieve a good balance between diversity and solution quality with such approaches.

**Deep Visual Constraints (DVC)** [83] is another closely related method, which learns shape-conditioned functional representations that represent an underlying kinematic constraint function. RPDiff can be interpreted as very similar to DVC if DVC was to be (i) extended to deal with unknown objects *and* scenes, (ii) simplified to avoid representing objects as neural field representations, and (iii) directly predicted gradients of the constraint function rather than representing the constraint function itself. More specifically, the update in each RPDiff iteration at test-time can be viewed as predicting the direction to move in, so as to come closer to minimizing/satisfying some underlying constraint function that encodes the geometric relation/constraint that should be satisfied. On the other hand, applying DVC to our scenario would involve the steps of predicting the constraint function value itself, and then performing either gradient-based or zero-order optimization to produce a solution that satisfies the learned constraint function. We did not implement or compare against this version of our approach, as we found satisfactory results using our method of directly predicting pose updates and directly encoding observable point clouds (rather than mapping to neural field representations), but it may be worth exploring the differences and tradeoffs between these two highly related yet subtly distinct approaches in the future.

**Relationship with energy-based learning**. Recently, many works have drawn relationships between the paradigms of energy-based modeling (EBM) [84, 85] and de-noising diffusion (e.g., [56, 57, 86–88]). As such, there have also been associated successes applying EBMs to object rearrangement in robotics (e.g., [89]). We highlight one core relationship between these two learning paradigms in our discussion on DVC above, which is that of prediction by explicitly optimizing a learned energy/cost/constraint function vs. training a diffusion model to directly approximate a gradient of such an underlying function. A common current intuition is that diffusion models have the advantage of being more stable and easy to train than EBMs [56], but there remains more work to be done in clarifying the degree to which this is true (i.e., see [86]) and exploring which among these closely related approaches is most suitable in specific robotics problems.

## A11   Model Architecture Diagrams

| Parameter | Value |
|---|---|
| Number of $\mathbf{P_O}$ and $\mathbf{P_S}$ points | 1024 |
| Batch size | 16 |
| Transformer encoder blocks | 4 |
| Transformer decoder blocks | 4 |
| Attention heads | 1 |
| Timestep position embedding | Sinusoidal |
| Transformer embedding dimension | 256 |
| Training iterations | 500k |
| Optimizer | AdamW |
| Learning rate | 1e-4 |
| Minimum learning rate | 1e-6 |
| Learning rate schedule | linear warmup, cosine decay |
| Warmup epochs | 50 |
| Optimizer momentum | $\beta_1 = 0.9$, $\beta_2 = 0.95$ |
| Weight decay | 0.1 |
| Maximum training timestep $T$ | 5 |
| Maximum $\mathbf{P_S}$ crop size $L_{\mathrm{max}}$ | $\mathbf{P_S}$ bounding box maximum extent |
| Minimum $\mathbf{P_S}$ crop size $L_{\mathrm{min}}$ | 18cm |

Table 4: Training hyperparameters

| Parameter | Value |
|---|---|
| Number of evaluation iterations $I$ | 50 |
| Number of parallel runs $K$ | 32 |
| Default value of $A$ in `i_to_t` | 10 |
| Expression for $p_{\mathrm{AnnealedRandSE(3)}}(\cdot \mid i)$ | $\mathcal{N}(\cdot \mid 0, \sigma(i))$ |
| $\sigma(i)$ in $p_{\mathrm{AnnealedRandSE(3)}}$ (for trans and rot) | $a * \exp(-bi/I)$ |
| Value of $a$ (axis-angle rotation, in degrees) | 20 |
| Value of $b$ (axis-angle rotation) | 6 |
| Value of $a$ (translation, in cm) | 3 |
| Value of $b$ (translation) | 6 |

Table 5: Evaluation hyperparameters

| Downsample point clouds | $(N+M) \times 3$ |
|---|---|
| One-hot concat | $(N+M) \times 5$ |
| Linear | $(N+M) \times d$ |
| Concat `pos_emb`$(t)$ | $(N+M+1) \times d$ |
| $\left[\quad\text{Self-attention (scene)}\quad\right]_{\times 4}$ | $M \times d$ |
| $\left[\begin{array}{c}\text{Self-attention (object)}\\\text{Cross-attention (object, scene)}\end{array}\right]_{\times 4}$ | $(N+1) \times d$ |
| Global Pooling | $d$ |
| Residual `pos_emb`$(t)$ | $d$ |
| MLP (translation) | $d \rightarrow 3$ |
| MLP $\rightarrow$ orthonormalize (rotation) | $d \rightarrow 6 \rightarrow 3 \times 3$ |

Table 6: Transformer architecture for predicting SE(3) transforms

| Downsample point clouds | $(N+M) \times 3$ |
|---|---|
| One-hot concat | $(N+M) \times 5$ |
| Linear | $(N+M) \times d$ |
| $\left[\quad\text{Self-attention (scene)}\quad\right]_{\times 4}$ | $M \times d$ |
| $\left[\begin{array}{c}\text{Self-attention (object)}\\\text{Cross-attention (object, scene)}\end{array}\right]_{\times 4}$ | $N \times d$ |
| Global Pooling | $d$ |
| MLP $\rightarrow$ sigmoid (success) | $d \rightarrow 1$ |

Table 7: Transformer architecture for predicting success likelihood

