# OpenReview forum: "Shelving, Stacking, Hanging: Relational Pose Diffusion for Multi-modal Rearrangement"
_robot-learning.org/CoRL/2023/Conference — CoRL 2023 Poster_

### Official Review · Reviewer_vuUj · 2023-06-28

**Confidence:** 2
**Originality:** Very Good
**Technical Quality:** Excellent
**Clarity Of Presentation:** Excellent
**Impact:** 4

**Recommendation:**

Strong Accept: I recommend accepting the paper and will argue for my recommendation even if other reviewers hold a different opinion.

**Review:**

The proposed approach addresses a complex task of multi-modal placing. The method is sound, and the technical specifications seem to be logic and reasonable. Their results illustrate the capabilities of the framework across a diverse range of rearrangement tasks involving objects and scenes that present a large number of feasible rearrangement solutions. Overall, the paper is well written, coving the relevant literature reviews with distinct demonstration of the merits of the proposed approach.

I recommend acceptance of this paper.


**Quality Of The Limitations Section:**

Limitations are addressed clearly

**Questions For Rebuttal:**

I do not have any specific questions.

**Robotics Focus:**

Highly relevant to robotics but no hardware experiments

**Summary Of Paper:**

This paper presents an approach for rearranging objects in a scene to achieve a desired placing relationship, operating with geometries, poses, and scene layouts. The system produces multi-modal distributions of object transformations for rearrangement. It is claimed that it can overcome the difficulty of fitting multi-modal demonstration datasets and facilitating integration with planning algorithms that require diverse actions to search through.

**Summary Of Recommendation:**

This is a paper highly relevant to robotics, with learning approaches for addressing a critical robotics problem. It is well developed, and I highly recommend it for acceptance.

---

> ### Author Response · Authors · 2023-08-10
> **Response to vuUj**
>
> Thank you for the encouraging review, we appreciate the positive comments. Please let us know if there are any remaining questions or concerns.

---

### Official Review · Reviewer_SEr7 · 2023-07-16

**Confidence:** 3
**Originality:** Good
**Technical Quality:** Very Good
**Clarity Of Presentation:** Very Good
**Impact:** 3

**Recommendation:**

Weak Accept: I recommend accepting the paper, but will not argue for my recommendation if the majority of other reviewers have a different opinion.

**Review:**

Strengths
* Novelty: The paper shares a similar spirit to recent works applying diffusion methods to robot manipulation ([41, 58]). This is a nice and simple formulation for leveraging diffusion models for manipulation.
* Notably, the main strength is to produce multimodal predictions.
* The authors demonstrate the ability to successfully execute their approach in the real world.
* Clarity: Overall the paper is clear.

Weaknesses
* Approach assumes that a point cloud segmentation is already provided for the scene and object (ie. the object is treated as one class, scene as another). It's not clear that this will always be easy to obtain
* Are there better alternatives to random perturbations in the forward diffusion process?
* Requires a manually defined offset based on prior knowledge of the task (line 198)
* Open loop execution

Clarity
* Page 2 describes a lot of motivation and overview of the pipeline. It would be useful to have associated images/figures throughout the explanation. For example:
    * Providing examples of the "local cropped region" around the object in line 67
    * What the "noised point cloud" looks like in line 51
    * What a "large perturbation" looks like in line 56
* The goal specification is not very clear in paragraph 1. Providing examples would help clarify things further. For example: for shelving books, is there a specific "shelving books dataset"? Is it one dataset per task, or can we use one dataset to learn multiple tasks? Is there a way to specify where to place the book, or is the goal to shelve a book in general? This is clarified only in the experiments section.

**Quality Of The Limitations Section:**

Limitations are addressed clearly

**Questions For Rebuttal:**

* How does this compare to other related works such as Diffusion Policy [58]? It would've been nice to compare against [41] as well. Even though [41] uses language input, the authors claim that the main bonus of the proposed approach is in its multimodal predictions. Is [41] able to achieve this just as well?
* How much do you add noise? e.g. Fig. 2b shows the book is very close to the shelf, but the initial starting pose in the supplementary demo is much farther away from the shelf.
* What is the real-world performance quantitatively? e.g. success rate
* How are the point clouds segmented for scene and object in the real world (line 206)?
* Fig. 3a: Why does recall terminate at ~0.68? Is it because with a large number of K samples, we don't get 1.0 recall? (this is fine, just asking for clarity)
* What are the main failure cases of the proposed approach?

**Robotics Focus:**

Sufficient demonstration on hardware

**Summary Of Paper:**

The authors propose an iterative diffusion method for object-scene re-arrangement. For tasks such as shelving books, the authors propose using a DNN to predict a transform a given object (e.g. book) from an initial pose to a final predicted pose, and execute this behavior in an open-loop manner. Specifically, the DNN predicts a series of transforms in an iterative manner, similar to diffusion models. The approach is able to produce multimodal predictions (e.g. multiple possible locations to shelve the book) and generalize to different environments and initial setups.

**Summary Of Recommendation:**

Overall, the paper is well written and presents a nice, simple formulation for applying diffusion models to robotics in order to achieve multimodal predictions. Moreover, the results are exciting, showing very high precision and recall in being able to achieve these multimodal predictions. The authors additionally provide real-world demonstrations. I'd be curious how the proposed method compares to recent papers such as [41, 58].

---

> ### Author Response · Authors · 2023-08-10
> **Response to SEr7 [Part 2/2]**
>
> > real-world performance quantitatively
>
>
> Unfortunately, we did not have time for quantitative eval of the real system, so our real-world results are limited to qualitative demos. Part of the challenge with setting up the real world evaluation is the need to make the surrounding system components as robust as possible, to allow large-scale evaluation without components that are unrelated to our core method causing execution failure. We observed a large fraction of failed execution attempts in the real world are caused by issues with inverse kinematics and collision-free trajectory execution, open loop execution (which assumes there is no in-hand object motion during/after grasping), imperfect computation of pre-placement offset poses, and slight reduction in placing prediction accuracy due to the distribution shift between simulated point clouds used for training and real world point clouds obtained with Realsense cameras. Our goal is to keep improving the full-stack system in the real world to enable more thorough real world evaluation in the future.
>
>
> > How are the point clouds segmented for scene and object in the real world
>
>
> We used Mask-RCNN together with the human operator clicking a pixel in the observed image indicating which mask corresponds to the manipulated object.
>
>
> > recall terminate at ~0.68
>
>
> Great observation. We find there remains some bias toward and/or away from certain modes, so with the current models + data + training scheme, we did not achieve complete coverage (note that coverage considers achieving both all possible positions *and* orientations).
> - This could possibly arise from bias/spurious correlations in the object point clouds and the demonstrated placements. For example, the initial book point clouds often had one large face visible with the other less visible, and there may have been some spurious correlations in the demonstrations showing the visible side aligned with the open slot in a particular orientation more frequently, causing this bias to be acquired by the model.
> - Another observation - occasionally, two modes are very close together (i.e., two open slots might be literally right next to each other). In these cases, the model may have again learned a biased solution either preferring one over the other, or reaching an average between these very-nearby solutions, leading the further-away/separate solution(s) as never being reached. This would be a very interesting detail to study further, we will explore it more if we have time before the final version.
>
>
>
>
> > main failure cases of the proposed approach
>
> We will expand discussion on primary failure modes in our updated paper. A few expanded details on failure cases in simulation are mentioned below.
>
> In simulation:
> - We discuss not achieving perfect coverage above
> - For success rates
>     - The model occasionally tries to place in parts of the scene that are cannot be reached (e.g., placing a book where there is already a book). Sometimes this is due to ambiguity in the scene object poses (i.e., it can be hard to distinguish the front vs. back of the shelf).
>     - It sometimes places an object “just off”, e.g., the handle of the mug is aligned to the rack, but isn’t far enough “on” the hook, so it just misses and doesn’t end up properly hanging. Similarly, sometimes the can placement is in collision with the stack of cans below.
>
> In the real world: you can see some of our discussion on this point in the response to HwxF (see “Real-world analysis/Sim2real gap”).
>
> [1] M. Janner, et al, “Planning with diffusion for flexible behavior synthesis”, ICML 2022
>
> [2] A. Ajay, et al, “Is Conditional Generative Modeling all you need for Decision-Making?”, ICLR 2023
>
> [3] J. Carvalho, et al, “Motion Planning Diffusion: Learning and Planning of Robot Motions with Diffusion Models”, arXiv 2023
>
> [4] W. Gao, et al,  “kPAM 2.0: Feedback Control for Category-Level Robotic Manipulation”, RA-L 2021
>
> [5] O. Spector, et al, “InsertionNet 2.0: Minimal Contact Multi-Step Insertion Using Multimodal Multiview Sensory Input”, ICRA 2022

---

> > ### Comment · Reviewer_SEr7 · 2023-08-12
> > **Response to rebuttal**
> >
> > After reading the rebuttal and other reviews, I maintain my rating (weak accept). Thank you to the authors for addressing my concerns, and please make sure to update the paper accordingly.

---

> ### Author Response · Authors · 2023-08-10
> **Response to SEr7 [Part 1/2]**
>
> > assumes that a point cloud segmentation is already provided for the scene and object
>
>
> This is a fair concern. We only require the *object* to be segmented from the observation (and assume it’s easy to remove background components, e.g., the table). We consider the whole rest of the environment observation to represent the “scene”. This is a limitation, but we argue its a reasonable assumption based on two observations:
> 1. Off-the-shelf image segmentation has continued to dramatically improve (i.e., SAM)
> 2. Even without perfect initial segmentation, our rearrangement problem requires first *grasping* an object to be placed. There are straightforward ways to obtain a refined + more complete object point cloud of the object once it’s held in the gripper.
>
>
> > alternatives to random perturbations in the forward diffusion
>
>
> Good question. We didn’t explore many other options for the forward noising process and found that our random perturbation scheme worked well, but given progress in the diffusion literature it’s possible that better methods exist. Feel free to elaborate if your concern isn’t properly addressed/you have more specific thoughts on improved ways to create the forward process.
>
> > Requires a manually defined offset based on prior knowledge of the task
>
>
> Agreed this is a limitation, we will update the limitations section to reflect this. Future versions ought to introduce predictions of more intermediate waypoints (note diffusion has shown to be useful in this context as well, e.g,. for motion planning/trajectory modeling [1, 2, 3]).
>
>
> > Open loop execution
>
>
> Agreed this is a limitation. Note it’s still rare to see closed-loop policies generalizing to the level we show and manipulating grasped objects to align with an external part of the environment. [4, 5] are inspiring examples that do perform closed-loop insertion/”tool use”/placing, perhaps there are lessons to be learned from these approaches in the future.
>
>
> > It would be useful to have associated images/figures throughout the explanation.
>
>
> We will take this suggestion into account and try improving the Fig. 2 quality and more explicitly connect the visual examples to the explanation in the writing.
>
>
> > The goal specification is not very clear
>
>
> Thanks for pointing this out, we will make this more clear.
> - There is a specific dataset for each task, and we trained one model on each respective dataset. Please see our response to ZYnf where we try training a multi-task model
> - Currently, our model aims to find *all* possible solutions. Similar to using our success classifier to rank and select a single prediction to execute, we could incorporate a “placement location cost function” that ranks the predicted poses relative to how close they are to a desired 3D position/region for placement.
>
>
> > How does this compare to other related works such as Diffusion Policy [58]? … [41]... Is [41] able to achieve [multi-modal predictions]?
>
>
> RE [41], please see our general response
> RE [58], they also motivate using diffusion due to its ability to handle multi-modality. Their emphasis is on closed-loop visuomotor policy learning, whereas we focus on predicting rearrangement transforms. They don’t show as much generalization across different objects/scene layouts, whereas all our test cases involve held-out object shapes and scenes with unseen layouts
>
>
> > How much do you add noise? e.g. Fig. 2b shows the book is very close to the shelf, but the starting pose in the demo is much farther away
>
>
> The object poses are initialized to be within the bounding box of the scene point cloud, with a random SO(3) rotation. While the starting poses of the real-world objects are further away, we can compute and apply a transform to the object point cloud which initializes its position in the bounding box of the scene point cloud (still with a random rotation).

---

### Official Review · Reviewer_HwxF · 2023-07-17

**Confidence:** 3
**Originality:** Good
**Technical Quality:** Good
**Clarity Of Presentation:** Good
**Impact:** 3

**Recommendation:**

Weak Accept: I recommend accepting the paper, but will not argue for my recommendation if the majority of other reviewers have a different opinion.

**Review:**

Quality:
Overall, the quality of the paper is high. The authors provide a clear and well-structured explanation of their proposed system for object rearrangement using 3D point clouds. They present a good enough problem analysis and provide fair descriptions of the methodology, experiments, and results. It’s hard to evaluate the technical foundation, mostly due to too much preprocessing / manual manipulation of the data (most of the details are included in the appendix). I understand that this is unavoidable due to the inherent requirements of the problem setup, but at times it feels like some critical details were omitted to push the reader away from the relatively ‘trickier’ parts of the approach that might indeed hinder the application of the approach in broader scenarios. But it definitely addresses important challenges in the field of robotics learning, so including a better discussion of those ‘hardships’/extra steps needed would also inform the reader about the complexity of the problem.

Clarity:
The paper is generally well-written and organized. The introduction effectively motivates the problem of object rearrangement and provides a clear context for the research. The methodology section describes the proposed system in a detailed and coherent manner. The experimental setup and evaluation metrics are clearly defined. The figures and captions are informative and support the text. However, in some sections, the paper could benefit from additional clarifications, especially regarding the specific implementation details.

Originality:
The paper presents a novel approach to object rearrangement by leveraging multi-modal predictions, iterative pose de-noising, and local scene conditioning. While there have been prior works on object rearrangement and pose estimation, the combination of these techniques in the proposed RPDiff system appears to be original. The authors contribute to the field by addressing challenges associated with multi-modality and generalization in object rearrangement tasks, and they introduce a new method that surpasses existing approaches in terms of precision and coverage.

Significance:
The significance of the work is fair,, specifically in the domain of object manipulation and cleanup tasks. The ability to rearrange objects in scenes accurately and efficiently is crucial for many real-world applications. The proposed RPDiff system tackles key challenges in this area, including handling multi-modality, generalizing to diverse scene layouts, and producing precise and accurate predictions. The experimental evaluations demonstrate the superiority of RPDiff over existing methods, highlighting its practical value and potential impact on robotics research and applications.

Strengths:

- Novel approach: The proposed Relational Pose Diffusion (RPDiff) system addresses the problem of object rearrangement in a scene using 3D point clouds, allowing it to handle unknown geometries in the real world.

- Multi-modal generation: RPDiff overcomes the challenge of fitting multi-modal demonstration data by using an iterative pose de-noising process, producing diverse and precise solutions for rearrangement tasks.

- Local conditioning: The system incorporates local conditioning by cropping a region near the object in the scene point cloud, improving generalization and precision by focusing on task-relevant details.

- Evaluation in simulation and real-world: The system is evaluated on three distinct rearrangement tasks in both simulation and the real world, demonstrating its effectiveness and applicability in different scenarios.

Weaknesses: (the authors acknowledge most of those points to some degree)

- Clarity: RPDiff employs a transformer architecture featuring self-attention. However, the authors state that they require an extra step called Local Conditioning, which involves cropping the scene around the object. The main text lacks clarity regarding the model's pipeline and requires further explanation. Consequently, it remains unclear why the additional Local Conditioning is necessary while using the self and cross attention mechanisms, as they should be capable of detecting relevant and irrelevant points in the scene.

- Data requirements: The system relies on scripted policies in simulation to obtain demonstration data, which may limit its applicability to new tasks.

- Real-world analysis: As the authors mentioned in their limitations, they had problems with switching from simulation to real-world experiments. Consequently, all their main results were obtained solely from simulations. In the Appendix, they acknowledged that they observed worse overall prediction performance in the real world, however they did not provide specific details about the performance.

- Sim2real gap: As highlighted by the authors as well, there is a potential sim2real gap due to training on simulated point clouds. Incorporating sim2real transfer techniques and finetuning on real-world data could improve the system's performance in real-world scenarios. However, the paper does not go into the details about how serious this gap was, how they addressed it, and it does not extensively address techniques to bridge this gap, leaving room for further investigation.

- Open-loop execution: The system executes the predicted placements in an open-loop manner, which may limit its ability to track progress and react to disturbances. As mentioned a closed-loop policy could enhance the system's performance in dynamic environments. However, how to integrate the whole approach into such a policy has not been discussed. For instance, considering the final phases of such rearrangement tasks (e.g., shelving a book between other books), the visual perception data may become less and less informative (e.g., due to occlusions).

- The paper assumes known object states and 3D models (at least in training as far as I understood), which may not be realistic in some real-world scenarios. The reliance on scripted policies for data generation limits scalability.

Minor comments:
- the term multi-modality sounds like sensor modalities. I don’t know whether this usage is well-accepted within this (object rearrangement) context but if not, then a different naming might be more informative / less confusing.

- Generalization: The paper should be more precise about what is meant by generalization in these scenarios, while being more clear about train and test data.

- line 115: identify -> identity

**Quality Of The Limitations Section:**

Additional details required

**Questions For Rebuttal:**

- Data preparation: For finding the goal configuration, how is the transformation computed given an initial object and scene point cloud data? There is some explanation in the Appendix, but I believe this information is important enough to be included in the main text, esp. to illustrate that there is a lot of domain knowledge incorporated into the data generation process.

- How viable pre-trained representations would be for overcoming the data generation challenges? How could the data requirements be reduced?

- As mentioned before, why is the additional Local Conditioning necessary while using the self and cross attention mechanisms?

- The paper assumes known object states and 3D models, which may not be applicable in all real-world situations. Could you discuss possible extensions or adaptations of RPDiff to handle scenarios where object states and models are uncertain or partially known?

- Could you provide more insights into the computational efficiency and real-time performance of RPDiff?

- The reliance on scripted policies for data generation raises concerns about scalability. Are there any plans to explore methods for collecting or generating data in a more automated or self-supervised manner? How might this impact the performance and generalization capabilities of RPDiff?


**Robotics Focus:**

Sufficient demonstration on hardware

**Summary Of Paper:**

The proposed system, RPDiff, addresses the problem of object rearrangement in a scene by operating directly on 3D point clouds. It overcomes challenges associated with multi-modality and generalization to diverse scene layouts. The main contributions of the paper are:

- Iterative Pose De-noising: RPDiff uses a diffusion model to perform iterative de-noising, gradually reversing the perturbation process applied to the object point cloud. This allows the system to generate diverse solutions for rearrangement tasks.

- Local Conditioning: To support generalization to novel scene layouts, RPDiff locally encodes the scene point cloud by cropping a region near the object. This allows the model to focus on task-relevant details and ignore irrelevant global structure, improving generalization.

Evaluation and Results: RPDiff is evaluated on three distinct rearrangement tasks in both simulation and the real world. It outperforms existing methods in terms of success rate and coverage.


**Summary Of Recommendation:**

The paper presents a well-structured demonstration of the work on using a diffusion model called RPDiff as a transformation prediction network in robotics. The authors draw from other domains and previous works to develop their approach. The writing is concise and easy to follow. However, there are some clarity issues, particularly regarding the pipeline and the necessity of local conditioning alongside self and cross attention mechanisms. The lack of real-world experimental analysis impairs the actual impact of the work. While the paper combines ideas from various domains, its main contribution seems to be the iterative use of the diffusion model, making it an incremental rather than groundbreaking work. Some important details are relegated to the appendix instead of being included in the main text.

---

> ### Author Response · Authors · 2023-08-10
> **Response to HwxF [Part 3/3]**
>
> > Real-world analysis/Sim2real gap
>
>
> Thanks for pointing this out. We will update the Appendix with a section on “real world performance analysis and sim2real challenges”.
>
>
> We can breakdown the performance analysis in terms of (1) placement prediction accuracy and (2) robot execution success.
>
>
> (1) Placement prediction accuracy
> - The distribution shift between simulated and real point clouds was the largest source of inaccurate transform predictions. This shift is in part caused by depth sensor noise, point cloud outliers, imperfect camera calibration, and shiny object surfaces.
> - Another source of distribution shift is “not perfectly realistic” scenes we generated and used for training. Sim2real pipelines are still bottlenecked by effort of creating realistic 3D assets *and* configuring diverse yet structurally/physically valid scene layouts to train on.
> - We explored some techniques to mitigate negative effects of sim2real gaps, such as adding small per-point Gaussian noise to the point clouds and simulating additional occlusions from randomly posed synthetic cameras. We observed some marginal benefit in applying these techniques but didn’t rigorously evaluate how much they helped.
> - Other ideas to explore for reducing sim2real gap with 3D point clouds:
>     - Using predicted depth models that can produce cleaner depth images than our depth sensors
>     - Using better depth sensors
>     - Training and deploying more high-fidelity noise models to augment the simulation data and make it more similar to the real world
>
>
> (2) Robot execution success
> - One factor is un-accounted object motion that occurrs while grasping/moving the object. We made the simplifying assumption that objects don’t move during grasping and that, after grasping, the object is rigidly coupled to the gripper. If there is object movement during grasping/trajectory execution, the placement may become inaccurate. One way to mitigate this would be to estimate object motion post-grasp and account for this when reaching the placing pose. We did not have time to implement this or similar system engineering improvements, but it should be straightforward to pursue in the future.
> - Another source of failure was imperfection in our computation of task-specific pre-placement offset poses. Noise in the point cloud (and limitations/non-robustness of the heuristics we used) meant sometimes our final pose predictions looked great but the pre-placement offset we obtained did not allow a feasible approach to reach the predicted pose. We definitely want to incorporate a prediction of more feasible waypoints along the *path* to the placement in future versions of the system.
>
>
> > more insights into the computational efficiency and real-time performance
>
>
> We timed the forward pass (with our hyperparameters) to require 280ms with K=32 and 49ms with K=1. We also measured the time required for 50 eval iterations - it took 3.3s with K=1 and 24.46s with K=32 (all on a V100 GPU). Multi-step inference is mainly bottlenecked by other operations like (re-)cropping the scene point cloud - our batched implementations of these can be improved to reduce runtime. We leave optimizing computational speed for real-time performance to future work and we will add discussion on computational efficiency in the Appendix.
>
>
> > hard to evaluate the technical foundation... too much preprocessing / manual manipulation of the data… critical details omitted…‘trickier’ parts [may] hinder the application of the approach in broader scenarios… including a better discussion of those ‘hardships’/extra steps needed would also inform the reader about the complexity of the problem
>
>
> We appreciate this feedback. It reveals that we both haven’t done a good job clarifying the core assumptions required by our method and haven’t painted a clear picture of the full system engineering we did to achieve our results (but which are not necessarily critical to the core components of the technical message we want to communicate). We will take this feedback into account in the revised version of our draft.
>
> [1] W. Gao, et al,  “kPAM 2.0: Feedback Control for Category-Level Robotic Manipulation”, RA-L 2021
>
> [2] O. Spector, et al, “InsertionNet 2.0: Minimal Contact Multi-Step Insertion Using Multimodal Multiview Sensory Input”, ICRA 2022
>
> [3] X. Yu, et al, “PointBERT: Pre-training 3D Point Cloud Transformers with Masked Point Modeling”, CVPR 2022

---

> > ### Comment · Reviewer_HwxF · 2023-08-11
> > **Rebuttal response**
> >
> > Thank you for the detailed clarifications, not just to my own comments but also to points raised by other reviewers as well. I will consider these explanations in my final evaluation.

---

> ### Author Response · Authors · 2023-08-10
> **Response to HwxF [Part 2/3]**
>
> > methods for collecting or generating data in a more automated or self-supervised manner
>
>
> Self-supervised data collection for tasks like these is a very interesting and worthwhile pursuit but will be challenging. Typically, some form of human supervision/labeling/demonstration/task-specific knowledge and engineering will be required to get initial learning “off the ground”. That could come in the form of demonstrations, scripted policies in simulation, or reward functions, but it’s hard to get away from any of such supervision entirely.
> - One could try to do RL. Potential challenges are (1) exploration is likely to be very challenging  - it will be hard to “accidentally” happen upon trajectories that achieve tasks like putting a book in a shelf, and (2) we will need a reward function, which will likely also some kind of privileged information.
> - We could try using our trained RPDiff models to operate in the real world to collect new rollouts, and then find a way to *score* the quality of the rollouts. For those that have high score/succeed at the task, we can add them back into the dataset and train on them some more. This requires a way to score the rollouts and the ability to obtain a performant-enough initial policy which succeeds some significant fraction of the time.
> - We can use the trained models to bootstrap an exploration policy and directly perform RL, by adding some noise/perturbations around the nominal behaviors for exploration. Again, this requires at least a reward function/model to score the quality of the trajectories/final states that are reached to perform RL
>
>
> Overall, we agree that data collection is a massive problem in robotics. Many of the best systems currently use some form of imitation learning rather than RL, and these will always require demos. Whether demos come directly from humans, scripted policies, or classical planning/control algorithms is a fair question to consider, but addressing this specific data source question is not what our current efforts in this paper attempt or claim to solve.
>
>
> > Open-loop execution - how to integrate the approach into a [closed-loop] policy
>
>
> Agreed, this is an important problem. We acknowledge this in our limitations section and leave the closed-loop rearrangement investigation for future work. Some ideas on incorporating this method with a closed-loop policy:
> - Our approach can likely be used to predict the final placing pose and to move the object “near” this pose open loop (perhaps with the extra estimation module to track small object motions that occur).
> - Once the object is close to the final placing region, we can turn on on a reactive closed-loop policy. This policy would need to have been trained to perform high-rate feedback control from RGB-D/point clouds/other modalities, for performing the “last few inches” of the placement. It remains to be seen how to set up the policy learning pipeline to train this “last few inches” controller (see [1, 2] for examples of prior work that may potentially be useful for methods toward achieving this)
> - RE “visual perception may be less informative”, we agree this is very possible. Such tasks might reveal a need for more dynamic feedback in the form of force/torque or tactile sensing. Indeed, prior work [1] that performs closed-loop insertion relied heavily on force/torque wrist sensing. However, we imagine visual feedback will still play a critical role in adjusting the object when placing to achieve a precise alignment with the scene.

---

> ### Author Response · Authors · 2023-08-10
> **Response to HwxF [Part 1/3]**
>
> > [why] local conditioning is necessary … self/cross attention mechanisms [should] be capable of detecting relevant/irrelevant points
>
> The Transformer SA/CA are capable of this to some extent, but they are still susceptible to assigning importance to faraway points (especially on unseen samples) if they are included in the input. Local cropping acts as a “hard” attention mechanism that *forces* the model not to consider points that are too far away.
>
> > relies on scripted policies in simulation to obtain demonstration data [i] which may limit its applicability to new tasks… [ii] assumes known object states/3D models [which] may not be realistic some real-world scenarios… [iii] how is the transformation computed given an initial object and scene point cloud data? … there is a lot of domain knowledge incorporated into the data generation process… [iv] plans to explore methods for collecting or generating data in a more automated or self-supervised manner
>
>
> Please see the general response regarding scripted policies/3D models/heuristics.
> - [i] New data generation policies would be needed for new tasks with our current pipeline, but we could equivalently collect demonstrations for the new tasks via teleoperation/kinesthetic teaching.
> - [ii (1/2)] We only use 3D models for obtaining training data in sim, and we only use sim for training data because it’s easier to scale it up with scripted policies. Ideally, we will require less data, so that it’s within reach to provide these demos directly in with real world objects
> - [ii (2/2)] Our approach is related to teacher-student training (i.e,. distilling from privileged information to models that operate from perception).
> - [iii] Domain knowledge was only incorporated to allow for an *automatic* data generation process, and could have been avoided if demos were collected via, e.g., teleop instead.
> - [iv] Self-supervised data collection for tasks like these is a very interesting and worthwhile pursuit, but will be challenging.
>
>
> We will elaborate on each of these aspects and details regarding our specific approach for generating demonstrations in the updated version of the paper.
>
>
> > how is the transformation computed given an initial object and scene point cloud data [during demo collection]
>
>
> Regarding questions regarding specific heuristics we used for data generation – we outline more details on each task below, and will update the Appendix to elaborate on these details more thoroughly.
> - **Book/Shelf Placing** To start, we manually generated the bookshelf 3D models, as shelves with books placed randomly among the available locations on each shelf. In creating these shelf models, we both placed books in some of the existing slot locations, and also recorded locations and orientations near each of the remaining open slots. These open slot poses were then used when generating object-scene point cloud demonstrations, so that we could directly obtain SE(3) transforms that move books from their initial poses into final poses within the shelf. The real world alternative would be to manually configure a diverse set of shelves and book placements within the shelves, and collect demonstrations of placing new books on these shelves.
> - **Can/Cabinet Stacking** Very similar to the approach for book/shelf placing, we programmatically generated the 3D scenes with cabinets and existing stacks of cans, and recorded poses of available poses either on top of the existing can stacks or in regions large enough for making new stacks. Again, we use this information in the data generation phase to directly transform new cans into configurations that are either “on” an existing stack or begin a new stack in an open area. The real world alternative would be to manually configure a diverse set of cabinets and can stacks within the cabinets, and collect demonstrations of placing/stacking new cans within these cabinets.
> - **Mug/Rack-multi Hanging** The mugs come from ShapeNet, where the 3D models for each category are all canonically aligned (i.e., the opening is aligned with the +y axis and the handle is aligned with the +z axis of the body frame). We use knowledge of these canonical poses to approximate the pose of the handle on each mug, and similarly use approximate knowledge of the hook on each rack to roughly align the mug handle with the hook of a rack. Finally, since these estimates are not perfect, we sample random perturbations of these poses until one is found that (1) does not cause collision between the objects and (2) leads to stable hanging of the mug on the rack when it falls under gravity. Similar to the above two tasks, with a few racks and mug instances, such demos could be collected in the real world without introducing any such assumptions.

---

### Official Review · Reviewer_AZ9E · 2023-07-19

**Confidence:** 2
**Originality:** Good
**Technical Quality:** Very Good
**Clarity Of Presentation:** Very Good
**Impact:** 2

**Recommendation:**

Weak Accept: I recommend accepting the paper, but will not argue for my recommendation if the majority of other reviewers have a different opinion.

**Review:**

I'm not familiar with research on rearrangement.

## Strength:
1. Reasonable approach and clear writing
1. Comprehensive experiments in both simulation and real-world
2. good results in both simulation and real-world.

## Weakness:

1. I'm not sure the necessity to use diffusion models in the setting, even though they are popular nowadays, What's the benefit of using diffusion models.
2. In Fig3 (b), the ablation shows that the success rate is sensitive to the crop rate. In my opinion, this somewhat indicates that the proposed method is not robust. Also, does the same cropping method is applied to the baselines as well?
3. I'm also not sure how the novelty compared to [41], which also uses the diffusion models and seems to handle a more challenging task.

## Some minor suggestions
1. in L105 $f_\theta$ should include the $pos\\_emb(t)$, as it's a input variable.


**Quality Of The Limitations Section:**

Limitations are addressed clearly

**Questions For Rebuttal:**

I would appreciate it if the authors can address my questions on weakness part, especially on why diffusion models are required for this task given that $h_\phi$ is required to score the prediction poses in addition to $f_\theta$

L92 mentions that grasp sampler, IK solver are required. Is it specially required for the proposed method, or is it a requirement for baselines as well?  I'm not familiar with this task. So the authors may disregard me if the question do not make sense.

Lastly, Equation (2) is not very clear to me:
$\hat T ^{(i-1)}$ and $\hat T ^{(i)}$ are not used? Do you mean $\hat P_O^{i-1} = \hat T ^{(i-1)}\hat P_O^{i}$


**Robotics Focus:**

Sufficient demonstration on hardware

**Summary Of Paper:**

The high-level goal of this work is to rearrange rigid objects to  location that satisfies geometric relationship with the scene.
The author propose to use diffusion method for iterative pose denoising.
Experiments are conducted in both simulation and real-world to demonstrate the effectiveness of the propose method.

**Summary Of Recommendation:**

I have never read paper on object rearrangement before. It's also the first time I review a paper using diffusion models, though I read some papers before.  I understand the high-level goal and approach of this work, but I'm not experienced enough to assess the practicality, effectiveness and novelty compared to previous methods.  I give a weak accept because the overall approach looks reasonable to me, the results  (especially the supplementary video) looks good, and the writing is  clear.

---

> ### Author Response · Authors · 2023-08-10
> **Response to AZ9E [Part 2/2]**
>
> > novelty compared to [41] [which also uses the diffusion models and seems to handle a more challenging task]
>
> Please see the general response for discussion on relationship to StructDiffusion.
>
> > why diffusion models [given] that h is required to score the prediction poses
>
> We mainly introduce the success classifier as a simple means to rank the *set* of predictions and select a *single* one to execute. Even without $h_{\phi}$, we would need some mechanism for selecting a single output (we tried uniformly sampling among the outputs instead of scoring, results are in the appendix, 88% average SR with success classifier, ~83% with uniform sampling - so the performance is still good, but decreases slightly). One way we could attempt to *generate* the poses in the first place, just using $h_{\phi}$ is to perform search/optimization with respect to the score, and select one with the highest score. This is reminiscent of energy-based modeling (discussed above).
>
> > grasp sampler, IK solver.... Is it required for the proposed method, or is it a requirement for baselines as well?
>
> These are required for all methods, given our setup. Our model + the baselines are design to predict a transformation of the object, without considering how the transformation will be achieved. The grasp sampling, IK, and motion planning are one simple instance of considering how to execute the predicted object transform, from the point of view of the robot (which is the only part of the environment we directly control).
>
> > Equation (2) is not very clear to me
>
> The point clouds that are updated on each step are used as input to the model on subsequent iterations. The transforms that we update on each step are used to eventually produce the final **transformation** output that we execute to the object (i.e., at the end of the iterations, this solution corresponds to the variable $\hat{\mathbf{T}}^{0}$, which we have been updating throughout the iterations via the transform-only part of Eq. 2). Please feel free to follow up on this point if things are still not clear and we will try to elaborate further.
>
> [1] J. Urian, et al, “SE(3)-DiffusionFields: Learning smooth cost functions for joint grasp and motion optimization through diffusion”, ICRA 2023
>
> [2] Y. Du, et al., “Reduce, Reuse, Recycle: Compositional generation with energy-based diffusion models and MCMC”, ICML 2023
>
> [3] C. Chi, et al, “Diffusion Policy: Visuomotor Policy Learning via Action Diffusion”, RSS 2023

---

> > ### Author Response · Authors · 2023-08-15
> > **Checking in before end of rebuttal period**
> >
> > We hope you have had a chance to view our responses to your questions, and wanted to double-check that you don’t have any additional comments to add in response to our rebuttal. We are happy to provide any further information for remaining questions/concerns.

---

> > > ### Comment · Reviewer_AZ9E · 2023-08-16
> > > **Thanks for the response**
> > >
> > > Thank you for your detailed response. It addresses most of my concerns.
> > >
> > > I plan to maintain a weak accept decision, primarily because I am not familiar with this topic. However, the work overall appears to be decent to me.

---

> ### Author Response · Authors · 2023-08-10
> **Response to AZ9E [Part 1/2]**
>
> > benefit of using diffusion models
>
> We discuss advantages of using diffusion over other methods that can also deal with some multi-modality in the baseline/experiments section. Here is a summary of some of these advantages:
> - Compared to (C-)VAEs, diffusion models are more flexible and expressive, partly because they aren’t constrained to match a simple prior (i.e., a unimodal Gaussian, as is common with VAEs). There may also be a benefit in the precision/”sharpness” that diffusion achieves by making multiple iterative predictions, whereas VAEs typically predict the output in a single step. The benefit of diffusion over VAEs has been similarly observed by experiments conducted in [1], which uses diffusion models for generating a diverse set of 6D grasp poses.
> - We compare to a method that discretizes the space and performs classification over the discretized space. These are well-suited for handling multi-modality, but generally introduce some discretization error/imprecision. Our diffusion model operates in the continuous space, and thus achieve higher precision (especially with rotations).
> - Another approach could be to train an energy-based model (EBM). This would involve training a NN to act as a “cost function” that outputs a high score when the object is far away from a correct placement, and a low score when the object is at a correct placement. Given this model ($E_{\theta}$) one can perform gradient descent, searching for a placement that minimizes the output of $E_{\theta}$.
>     - This is highly related to diffusion models - diffusion can be considered as directly approximating the gradient of an underlying cost function, rather than representing the cost itself and taking an explicit gradient [2]. One main difference among these options is diffusion models are more stable to train than EBMs, i.e., as noted in the recent work [3])
>
> > success rate is sensitive to the crop rate… indicates that the proposed method is not robust
>
> This is a fair concern. The size of the crop and crop size reduction rate are hyperparameters. We did not exhausitevely search for the best value or evaluate the performance variation with different fixed crop sizes, but we obtained satisfactory results without any hyperparameter search when using a variable crop size (shown in the “Varying” row of Fig. 3b). We approximated the minimum crop size (used for both the “fixed” crop and the final size for “varying” crop) based on an approximate minimum bounding box that captures the point cloud of our largest object across tasks (the books) and kept this constant across all tasks and all experiments. Any smaller, and there might not be enough of the surrounding scene context that is used for prediction for some of the objects, and we did not want to add complexity by having per-task crop sizes (although this can probably be done to further improve performance).
>
> > does the same cropping method is applied to the baselines as well?
> - We use the same cropping method for the C2F-Qattn method. This approach first predicts a coarse object transform, applies this transform to the object point cloud, and crops the scene near the transformed object to make a subsequent “refinement” prediction.
> - For the NSM/CVAE-NSM methods, because these methods make the prediction in a single step, we cannot use any cropping method for them. Part of the purpose of comparing to these is to highlight that overcoming the problems of multi-modality/diverse predictions and generalization is much more difficult when considering the entire global scene.
> - Similarly, for the R-NDF baseline, we did not perform any cropping, highlighting that training an NDF representation on entire scenes leads to poor correspondence matching and inference of task-relevant object/scene parts that should be aligned for rearrangement. It’s possible that introducing cropping to the NDF representation would also help this baseline perform better, but this would involve a non-trivial exploration of how to modify the way the NDF models operate and what data they are trained on. This was outside the scope of this current work, but is a good idea for a complementary investigation.

---

### Official Review · Reviewer_ZYnf · 2023-08-02

**Confidence:** 4
**Originality:** Good
**Technical Quality:** Good
**Clarity Of Presentation:** Good
**Impact:** 3

**Recommendation:**

Weak Accept: I recommend accepting the paper, but will not argue for my recommendation if the majority of other reviewers have a different opinion.

**Review:**

In general, the paper presents a novel idea of using diffusion models to solve robotic manipulation tasks.
The paper contains a few interesting technical details, such as cropping the scene point cloud to focus on relevant parts, the loss function loss, the exact diffusion setup adjusted to predicting rigid transformations.
The reviewer appreciates the detailed appendix which clarified a lot of my initial questions.


However, I think that the paper is missing central arguments why the proposed method advances the state-of-the-art:

* The main weakness of the paper is the following: The data to train the models is generated in simulation using manually designed heuristics. While this in principle is sound and a common approach, in my opinion, one then has to show that the learned method exhibits capabilities that go beyond those manually designed heuristics. I currently do not see why the point clouds as obtained in this work cannot be transformed with the same heuristics as for generating the data.

* How does this method extend to multi-step manipulation problems?

* I suggest looking into these papers:
[1] Yifan You, Lin Shao, Toki Migimatsu, Jeannette Bohg. "OmniHang: Learning to Hang Arbitrary Objects using Contact Point Correspondences and Neural Collision Estimation". ICRA 2021.
[2] Jung-Su Ha, Danny Driess, Marc Toussaint. "Deep Visual Constraints: Neural Implicit Models for Manipulation Planning From Visual Input". RA-L 2022.
I suggest to potentially adding them as a baseline (especially OmniHang). There are also other papers exploring energy-based methods to deal with multimodality.

* Regarding the limitation section, the fact that there is no transfer to new tasks and that separate models for each task are trained could be added.

**Quality Of The Limitations Section:**

Additional details required

**Questions For Rebuttal:**

See review.

Additional questions:
- The authors state that the method generalizes over shapes. Where is this evaluated?
- In the Neural Descriptor Fields paper I see success rates of nearly 90% for hanging a mug, why is this baseline significantly lower here?
- I am a bit confused about the one-hot encoding in the transformer input. Is the reason to use this in order to share the point-cloud encoder between the scene and object point-clouds?
- "In both simulation and the real world, we manually completed offset poses for moving the object before executing the final placement.". Can this be explained in more detail?
- Are there ablations regarding the different loss terms?
- What about just learning the classifier and then use sampling/gradient descent?

**Robotics Focus:**

Sufficient demonstration on hardware

**Summary Of Paper:**

This paper proposes to use diffusion models to predict rigid transformations applied to object point clouds in order to solve a manipulation task. The model gets as input 3D point clouds of the object that should be manipulated and the surrounding scene. The diffusion model iteratively outputs transformations to rearrange the object to a goal configuration. A key aspect of the approach is that multiple solutions can be predicted. The training data is generated in simulation using manually designed heuristics.

**Summary Of Recommendation:**

As written in the review above, the paper contains novel ideas and is well written.

---

> ### Author Response · Authors · 2023-08-10
> **Response to ZYnf [Part 2/2]**
>
> > Can [manually computed offset poses] be explained in more detail?
>
> RPDiff predicts the final pose for rearrangement, but to *reach* this final pose, we still need a full feasible path of the object (and the robot, in the real world). We decided that solving the full collision-free path planning problem was beyond scope of the problem addressed in our work. Instead, we again used some heuristics for combining the predicted pose with some knowledge about the task to obtain an “offset pose”, which is the same as the final pose but translated by a some 3D vector (the heuristics are used to obtain the specific 3D vector). We then directly move to this “offset pose” without considering collisions and then actuate the object in a straight line along the 3D offset vector to attempt reaching the final pose. The “offset pose” and the heuristics we utilize are formed such that, if the final predicted pose is correct, then (1) moving to the offset pose, followed by (2) moving linearly along the offset vector, should lead to successful placement.
>
> We will update the paper to make these components more clear. We acknowledge that extending the method to predict the offset poses (and other feasible waypoints) would be ideal, but this was beyond the scope of the task we considered for this version.
>
> > What about just learning the classifier and then use sampling/gradient descent?
>
> This has been proposed and explored before, such as in [2] (for refinement of a grasp pose obtained from a C-VAE). [3] compared against this as a baseline and observed a difficulty in using it to obtain good results (perhaps due to regions where the classifier produces either very little or very noisy gradients) and that generating 6-DoF poses with diffusion performed better. Training such a model to operate through the whole the space becomes reminiscent of energy-based modeling, which can also be attempted and has strong relationships with diffusion [1, 3, 4], but it has been observed that EBMs can be harder to train (see [1, 5]).
>
> [1] C. Chi, et al, “Diffusion Policy: Visuomotor Policy Learning via Action Diffusion”, RSS 2023
>
> [2] A. Mousavian, et al, “6-DoF GraspNet: Variational Grasp Generation for Object Manipulation”, ICCV 2019
>
> [3] J. Urian, et al, “SE(3)-DiffusionFields: Learning smooth cost functions for joint grasp and motion optimization through diffusion”, ICRA 2023
>
> [4] Y. Du, et al., “Reduce, Reuse, Recycle: Compositional generation with energy-based diffusion models and MCMC”, ICML 2023
>
> [5] D.N. Ta, et al, “Conditional energy-based models for implicit policies: The gap between theory and practice”, arXiv 2022

---

> ### Author Response · Authors · 2023-08-10
> **Relationship to prior work (OmniHang, Deep Visual Constraints) [Part 2/2]**
>
> RE Deep Visual Constraints (DVC): We realized we can interpret RPDiff as very similar to DVC, if DVC was to be (i) extended to deal with unknown object *and* scenes and (ii) simplified to avoid representing objects as neural field representations. Details on this are elaborated:
> - We first note their setting involves representing a single “unknown” object. In contrast, we have one “unknown object” and an *entirely separate* “unknown scene”.
> - With this in mind, let’s examine one way to directly transfer DVC to the object-scene case:
>     - First - we can encode the scene into an implicit representation that must be evaluated at some 3D query points.
>     - Since the object being rearranged is also unknown, the query points would also have to be *predicted* (whereas in DVC, they are manually set + fixed on a static object, i.e., the gripper or the hook of the rack).
>     - The model would be trained to output the distance between these query points and their correct “final” configuration that is shown in the demonstrations (i.e., the model outputs 0 when the object is correctly posed, and some value > 0 when the object pose is perturbed, where the value is proportional to how far away the object is from the correct pose).
>     - At test time, we would (i) predict the query points and (ii) search for object transforms such that the NN model (when evaluated at the query points) output equals/approaches 0. Alternatively, we can directly perform something akin to gradient descent on the output of this function (since we can take the gradient of the NN output with respect to the object pose).
> - With this candidate pipeline for applying DVC to our setting in mind, the key changes we have in our method are:
>     - We don’t predict an implicit scene representation. Instead, we operate directly on the scene point cloud (or a crop of it) via self/cross attention
>     - We don’t predict a specific subset of object points that are important for interaction (i.e,. as query points). Instead, we also just operate on the whole observable object
>     - We don’t predict a distance to the correct pose and then search/optimize to minimize it. Instead, we directly predict the *direction* to move the object that *would take a step toward minimizing* such an underlying distance function. This is analogous to how, instead of optimization via gradient descent on a learned energy function, we can train a diffusion model to directly approximate this gradient.
> - Overall, this makes the core principles of our approach very similar: Our update on each iteration can be viewed as predicting the direction to move in, so as to come closer to minimizing/satisfying some underlying constraint function that encodes the geometric relation/constraint that should be satisfied
>     - Further, as described above, our method enables the overall approach to be simpler (no need to predict query points) and uses the variant of these techniques that has been found to train more stably (i.e., as discussed in [2], diffusion models have been found to train with greater stability than energy-based models).
>
> RE “other energy-based methods to deal with multi-modality”: As mentioned above, there are close relationships between diffusion-based inference and methods that optimize a learned energy function. However, it’s been observed that diffusion models are more stable and easy to train than energy-based models, so we did not explore comparing against the energy-based variant of our approach.
>
> [1] A. Mousavian, et al, “6-DoF GraspNet: Variational Grasp Generation for Object Manipulation”, ICCV 2019
>
> [2] C. Chi, et al, “Diffusion Policy: Visuomotor Policy Learning via Action Diffusion”, RSS 2023

---

> ### Author Response · Authors · 2023-08-10
> **Relationship to prior work (OmniHang, Deep Visual Constraints) [Part 1/2]**
>
> These are great suggestions. We should have cited and elaborated on the relationship between our method and these papers and will do so in the updated version.
>
> RE OmniHang: This method is composed of multiple stages: (1) coarse hanging pose regression, (2) keypoint detection + alignment, (3) CEM refinement with a learned reward function:
> - (Stage 1) This is analogous to our NSM + CVAE baseline, which we found performs much worse due to limits of predicting the full transform in a single step and limiting the latent variable noise to be drawn from a unimodal Gaussian prior.
> - (Stage 2) This is analogous to the R-NDF baseline, that finds task-relevant local geometry on the object/scene and brings them into alignment. One difference b/w OmniHang and R-NDF is R-NDF detection via nearest neighbor in a self-supervised feature space, while OmniHang trains a fully-supervised keypoint detector .
>     - We could try a keypoint detection model and use this to refine the predictions made by our method/NSM+CVAE/etc.. However, this introduces the drawback of requiring extra keypoint labeling to train the keypoint detector. These labels are easier to come by when operating with shapes in simulation, but if we move to using demonstrations collected directly in the real world, then the keypoints have to be labeled by hand. This is more laborious than just providing the demo for the rearrangement task.
>     - Futhermore, our method produces satisfactory results by directly (iteratively) regressing the rearrangement transform, so introducing this extra component has the extra drawback of adding un-needed complexity.
> - (Stage 3) This is similar to if we used our success classifier to refine the prediction by performing local optimization (i.e., to maximize the score of the classifier). This has also been deployed in other work for 6-DoF grasp generation [1]. Again, we could have introduced this as a way to refine the predictions, but we already obtained good results by directly refining the pose using our diffusion model.
>     - Another potential disadvantage of local refinement is the need to perform either (i) more expensive exhaustive local search of a refined pose, or (ii) to directly optimize the output of the classifier, and option (ii) can be susceptible to exploiting regions of space where the classifier outputs erroneously large scores, thus needing to add extra components that constrain the search to a local region and avoid optimal regions that exist outside the distribution the model was trained on.

---

### Author Response · Authors · 2023-08-10
**General Response**

We are grateful for the constructive feedback provided by the reviewers. The reviews agreed that our paper addresses an important problem (HwxF said we address “important challenges in the field”, vuUj said we study a “critical” problem that is “highly relevant to robotics”) and highlighted the quality of our results (AZ9E says “good results in simulation and real world”, SEr7 says our “results are exciting”). ZYnf also noted our method contains “novel ideas”, and SEr7 stated that we present a “nice and simple formulation”. Finally, the reviewers agree that our paper is well-written and clear.

We hope the reviewers find our response satisfactory. We would appreciate it if they could let us know if some concerns still remain.

**Summary of experiments/figures/material to be updated**
- New results on multi-step tasks
- New results showing ability to train a multi-task model
- Simplified loss and point cloud processing (based on suggested ablations)
- Expanded related work discussion
- Updates to pipeline figure
- Expanded discussion on:
    - Clarifying why we use scripted policies/task-specific heuristics/3D models in demos
    - Heuristics for generating demo transforms
    - Performance analysis + common failure modes
    - Sim2real challenges
    - Computing pre-placement offset poses
    - Computational efficiency

**Response to concerns about 3D models, scripted policies, manual heuristics (ZYnf, HwxF)**

The issue of data collection via scripted policies using heuristics and 3D models was mentioned by ZYnf and HwxF. We want to clarify a potential confusion that our method requires such assumptions. Our approach does *not* fundamentally rely on access to heuristics or privileged information. We only rely on demonstrations of object-scene point clouds in final configurations that satisfy the desired object-scene relation.
- The demonstrations could equivalently come from the real world (i.e., via teleoperation or kinesthetic teaching, similar to [1, 2]), but we were unable to collect a large number of these demonstrations in the real world, and instead opted to automatically collect them in sim.
- With this goal, we take advantage of privileged sim/3D model information and task-specific heuristics purely for automating data generation. This information includes 3D models, simulator states, canonical object poses, and available placing locations (which were recorded when scene objects were generated).
- One goal of future work is to bring down the num. of demos required so the training examples can be directly shown in the real world on real objects

RE ZYnf’s concern of showing “capabilities that go beyond those manually designed heuristics” -- the information used to generate the demos is *not* available in the real world. We only use the heuristics + privileged information to generate training data, and then obtain models that operate with point clouds, which *can* be deployed in the real world on unknown objects and scenes. This is analogous to the paradigm of teacher-student training in RL and imitation learning, wherein a “teacher” policy is trained using privileged information, and then distilled to a “student” policy that operates from perception [3, 4].

[1] C. Pan, et al, “TAX-Pose”, CoRL 2022

[2] A. Simeonov, et al, “Relational Rearrangement with Neural Descriptor Fields”, CoRL 2022

[3] T. Chen, et al, “System for General In-Hand Object Re-Orientation”, CoRL 2021

[4] A. Kumar, et al, “RMA”, RSS 2021

**Response to concerns about relationship to StructDiffusion (AZ9E, SEr7)**

We acknowledge the similarities between our method and [41]. The main difference between the works is the emphasis on different kinds of tasks and abilities
- As [41] also uses diffusion, they can also produce diverse outputs, but they don’t draw significant attention to this aspect. In contrast, we go so far as to quantify the coverage achieved by our approach, and we specifically highlight the diverse solutions our method achieves in both sim and the real world.
- Further, our local conditioning is more well-suited to rearranging single objects one at a time. It would be less natural for [41] to incorporate local scene conditioning because the global scene context is required to ensure their high-level abstract concept is being achieved.
- The transforms we execute in our results involve large 3D rotations and relatively precise alignment with the surrounding scene geometry, whereas the real-world demos shown in [41] involved smaller (mostly SE(2)) transformations. It’s likely [41] can also produce large object transformations and precise alignment, but this is one more example highlighting our focus on different aspects of rearrangement.
- Finally, we note [41] was only on arXiv at the time of submission, and it has since only been officially published in the RSS proceedings last month. We hope the reviewers and AC can acknowledge that our methods were developed at least somewhat concurrently.

---

### Decision · Program_Chairs · 2023-08-30

**Decision:**

Accept (Poster)

**Comment:**

The reviewers agree that this paper addresses an important problem and that the proposed approach, while rather incremental, is sufficiently novel. While several questions about technical description and experimental evaluation have been addressed by the author rebuttal and discussion, some concerns such as about the limited transferability to real-world remain.

The suggestions of the reviewers should be thoroughly incorporated in a camera-ready version of the paper.